# A MULTI-REGION BRAIN MODEL TO ELUCIDATE THE ROLE OF HIPPOCAMPUS IN SPATIALLY EMBEDDED DECISION-MAKING TASKS

## ABSTRACT

We present a multi-region brain model exploring the role of structured memory circuits in spatially embedded decision-making tasks. We simulate decision-making processes that involve the cognitive maps formed within the CA1 region of the hippocampus during an evidence integration task, which animals learn through reinforcement learning (RL). Our model integrates a bipartite memory scaffold architecture that incorporates grid and place cells of the entorhinal cortex and hippocampus, with an action-selecting recurrent neural network (RNN) that integrates hippocampal representations. Through RL-based simulations, we demonstrate that joint encoding of position and evidence within medial entorhinal cortex, along with sensory projection to hippocampus, replicates experimentally observed place cell representations and promotes rapid learning and efficient spatial navigation relative to alternative circuits. Our findings predict conjunctive spatial and evidence tuning in grid cells, in addition to hippocampus, as essential for decision-making in space.

## 1 INTRODUCTION

The hippocampus (HPC) plays a critical role in spatial, contextual, and associative learning and memory (O'Keefe, 1978; Dostrovsky & O'Keefe, 1971; Squire, 1992; Scoville & Milner, 1957). Cortical and subcortical regions are critical for tasks involving evidence accumulation and decision making (Pinto et al., 2019; IBL et al., 2023). Recent experiments examined the interplay of neural representations in these regions during the accumulating tower task that combined both spatial navigation and decision-making (Pinto et al., 2019; 2022; Nieh et al., 2021).

The task is as follows: As mice run through a virtual corridor, they are stochastically presented with towers on both sides (Nieh et al., 2021). At the end of the corridor, they must make a left (right) turn if there were more towers on the corresponding side. The task involves integrating evidence: deciding whether to turn right or left can be solved by taking the difference of the integrated number of towers on each side ("accumulated evidence"). Though the spatial locations of towers along each side are irrelevant to the task, neurons in area CA1 of the dorsal hippocampus signaled spatial location during the task with their place fields. Moreover, these fields formed combined cognitive maps of space and accumulated evidence, with individual place fields tuned to a specific combination of accumulated evidence and location.

This observation raises the question of why the hippocampus (HPC) is involved in evidence representation in the task: Does it play an important functional role in spatially embedded decision-making tasks, even when it does not seem that spatial information is necessary? And why are spatial and evidence represented together?

At the same time, understanding how different brain regions interact to solve spatially embedded decision-making tasks can shed light on how the brain distributes and coordinates computations across numerous substructures to allow it to flexibly and efficiently solve various complex tasks. Thus, the questions above also provide an opportunity to gain insights into the multi-region interactions that underpin complex cognitive functions. Such an understanding could reveal how cognitive maps could be leveraged not only to navigate physical spaces but also to guide cognitive decisions.

Decision-making tasks are often reinforcement learning problems (Gershman & Niv, 2015; Gershman & Daw, 2017), and various reinforcement learning (RL) models from formal Bayesian solutions

to deep reinforcement learning, have elucidated the demands and representational requirements of solving such tasks. The accumulating tower task has also been modeled with deep RL (Mochizuki-Freeman et al., 2023; Lee et al., 2024). What is mostly lacking in all these approaches, however, is the use of structured architectures and dynamics found in the brain and modeling the multi-system computations distributed across brain regions to solve RL tasks robustly and learn them efficiently.

Here, we developed a multi-region brain model that incorporates an architecturally and dynamically prestructured hippocampal-entorhinal circuit based on the memory model, Vector Hippocampal Scaffolded Heteroassociative Memory (Vector-HaSH) (Sharma et al., 2022; Chandra et al., 2023). The circuit interacts with cortical and sub-cortical regions that play the role of a decision-making actor, abstracted as a recurrent neural network (RNN) (Elman, 1990). Thus, the model generalizes Vector-HaSH into an RL problem solver with the addition of an actor/decision-making network. This is related to achieving autonomous machine intelligence positioned and discussed by LeCun (2022); we demonstrated, as a proof of concept, that sample-efficient learning in RL is achievable through an external content-addressable associative memory with a structured aspect. In particular, our work shows a structured conjunctive coding scheme (*i.e.*, grid cells as a canonical example drawn from biological representation learning) is a necessary structural representation for forming cognitive maps (world models), enabling individuals to learn quickly and navigate spaces efficiently.

We apply the model to test the counterfactual of agent reinforcement learning behaviors on the accumulating tower task, a spatially embedded decision-making task experimentally studied in rodents, assuming different inputs into grid and place cells. In these different scenarios, our model makes predictions about neural representations, the role of the hippocampal-entorhinal network in spatially embedded decision-making, and the efficacy of task learning.

The contribution of this paper is four-fold [1]:

- We propose and demonstrate a multi-region brain modeling framework that elucidates the underlying neural mechanisms and computational roles of entorhinal-hippocampal-neocortical interactions during spatially embedded decision-making tasks.
- The model permits a systematic study of how cognitive capabilities such as rapid learning and efficient navigation might emerge, and can guide and help interpret future experiments.
- We predict conjunctive location-evidence tuning in grid cells. We also predict that grid cell conjunctive tuning and entorhinal non-grid inputs to hippocampus are both necessary for the emergence of conjunctive hippocampal cognitive maps (Nieh et al., 2021).
- Finally, we show that the two properties above (conjunctive grid cell tuning and non-grid inputs to hippocampus) are essential for rapid learning and efficient navigation.

## 2 RELATED WORKS

### 2.1 BIOLOGICAL FOUNDATIONS OF HIPPOCAMPUS AND ENTORHINAL CORTEX

Hippocampal place cells, whose discharges represent locations in space (Dostrovsky & O'Keefe, 1971), form the foundation of the cognitive map theory (O'Keefe, 1978; Moser & Moser, 2016; Fenton, 2015). This theory provides an important framework for understanding how flexible and intelligent behaviors emerge from neuronal populations (Fenton, 2024). The cognitive map enables individuals to navigate spaces flexibly and organize memories, which are both investigated in this paper through our multi-region model, as well as construct coherent personal narratives of their experiences (O'Keefe, 1978; Tolman, 1948; Whittington et al., 2022; Fenton, 2024). Additionally, the discovery that the hippocampus encodes internal cognitive variables (Bostock et al., 1991; Ólafsdóttir et al., 2015; Nieh et al., 2021; Tavares et al., 2015) positions the place cell phenomenon as a model system for studying internally generated cognition. Moreover, interactions between the hippocampus and entorhinal cortex are well-established to support both navigation (McNaughton et al., 1996; O'Keefe, 1978) and declarative memory (Squire, 1992; Scoville & Milner, 1957). Grid cells (Hafting et al., 2005), located in the medial entorhinal cortex, provide a spatial metric through firing fields that form a periodic hexagonal pattern across the environments (Krupic et al., 2012). To reconcile this foundational understanding with experimental data, recent hypotheses propose that mechanisms underlying memory and planning have evolved from physical navigation processes,

---

[1]All code for running the model and its variants will be made available (upon publication) for reproducibility.

suggesting that neuronal algorithms for navigating both physical and mental spaces are inherently the same (Buzsáki & Moser, 2013). Combined with the established hippocampal-prefrontal functional interactions (Preston & Eichenbaum, 2013; Eichenbaum, 2017), our multi-region model is at the core of understanding the neural mechanisms underlying cognition. Recent advances in experimental techniques allow for the simultaneous recording of thousands of neurons (Jun et al., 2017; Steinmetz et al., 2021), enabling investigatations of how neurons across multiple brain regions coordinate to function coherently (Bondy et al., 2024). Our theoretical work, informed by experiments, bridges the gap with experimental progress, offering interpretable, mechanistic models to dissect the roles and interactions of individual regions in cognitive processes, thereby guiding future experiments.

## 2.2 Computational Models of Hippocampus and Entorhinal Cortex

The most prominent modeling approach to the entorhinal-hippocampal interaction includes the Tolman-Eichenbaum machine (Whittington et al., 2020), a statistical generative model, and Vector-HaSH (Chandra et al., 2023), a mechanistic model of neural circuits, grounded in neurobiological principles of connectivity, structure, and learning rules. Thus, Vector-HaSH provides an strong foundation for our work, facilitating a close, reciprocal relationship between theoretical predictions and experimental results. More generally, several data-driven approaches have been developed to understand multi-region interactions. Perich et al. (2020) provided an approach for inferring brain-wide interactions using data-constrained RNNs that directly reproduce experimentally obtained neural data. Koukuntla et al. (2024) developed a novel unsupervised machine learning method to disentangle low-dimensional shared and private latent variables across brain regions. Our mechanistic modeling approach brings a new perspective by integrating and unifying these statistical discoveries.

## 2.3 Bidirectional Insights Between Deep Learning and Neuroscience

Deep learning-based frameworks are widely used in systems neuroscience (Richards et al., 2019). For example, deep neural networks have emerged as plausible models of the brain (Sacramento et al., 2018; Whittington & Bogacz, 2017; Kumar et al., 2021), mimicking representational transformations in primate perceptual systems (Kell et al., 2018; Bashivan et al., 2019). Moreover, these models exhibit classic behavioral and neurophysiological phenomena when trained on tasks similar to those performed by animals (Banino et al., 2018; Pospisil et al., 2018; Wang et al., 2018). On the other hand, Yamins & DiCarlo (2016) measure the correlation between artificial neural networks (ANNs) and neuronal activities in the monkey visual cortex during image classification tasks, shedding light on the design of brain-like ANNs (Kubilius et al., 2019; Zhuang et al., 2021). Insights from brain-like models are just as valuable to the machine learning community as they are to neuroscience. For example, navigation is fundamental for humans but challenging for ANNs (Mirowski et al., 2017). Banino et al. (2018) leverage the computational functions of grid cells, critical for navigation in primates, to develop a deep RL agent with mammal-like navigational abilities.

## 3 Methods

### 3.1 Multi-Region Brain Model for Entorhinal-Hippocampal-Cortical Computation and Interaction

Our multi-region brain model integrates a cortical circuit modeled by an action-selection RNN policy, and an entorhinal-hippocampal circuit based on Chandra et al. (2023) that involves bidirectional computations between grid cells and place cells to associate, encode, and learn information from the environment. As described in Chandra et al. (2023) and shown in Fig 1A (purple and orange), the entorhinal-hippocampal memory scaffold is a bipartite architecture that comprises hidden (hippocampal) and label (grid cell) layers with computation based on known and inferred recurrent connectivity between entorhinal cortex and hippocampus (Witter & Groenewegen, 1984; Amaral & Witter, 1989; Witter & Amaral, 1991; Witter et al., 2017) and among grid cells in the entorhinal cortex (Burak & Fiete, 2009). The connections from grid cells to hippocampus, $\mathbf{W}_{pg}$, are set as fixed and random, while the connections from hippocampus to grid cells, $\mathbf{W}_{gp}$, are set once through associative learning and held fixed. The connections between hippocampus and non-grid EC, $\mathbf{W}_{ps}$ and $\mathbf{W}_{sp}$, are learned bidirectionally through associative learning. The grid cell layer is a $k$-hot modular vector imposed by local recurrent inhibition, where $k$ reflects the number of distinct one-hot modules. Each module has a unique period, and velocity inputs (*e.g.*, position and evidence) advance the phase of each module along its 2D representational space (*i.e.*, a 2D torus).

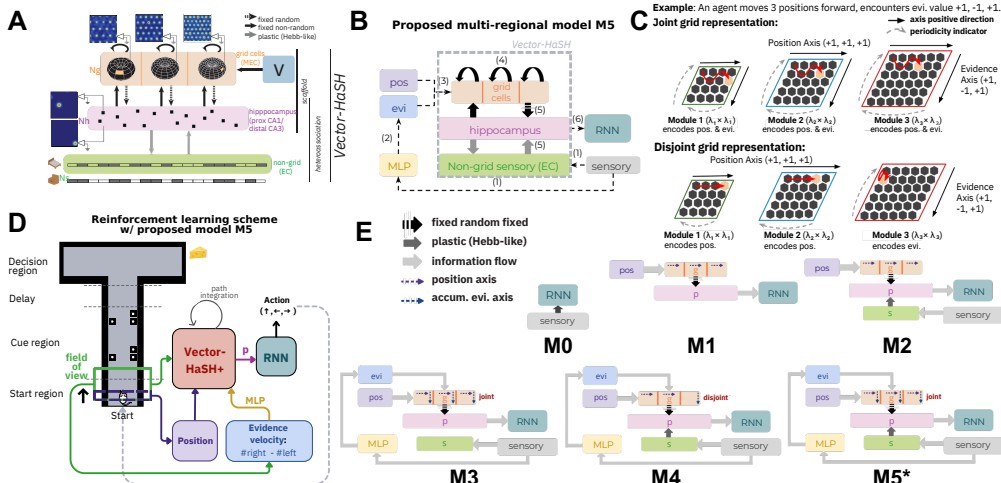

Figure 1: **Model and task schematics. (A)** Schematic of Vector-HaSH (Chandra et al., 2023), which provides the basis of our entorhinal-hippocampal circuit. **(B)** Schematic of the Vector-HaSH+ circuit, which we propose to model and investigate the neural computation process for spatially embedded decision-making tasks. The numbers in parentheses are the order of computation. **(C)** Schematic of grid cell coding, for a specific example of an agent moving 3 positions forward while encountering evidence value +1, -1, and +1. Here we assume the grid state is initialized at the top left corner, but this coding scheme is invariant regardless of the initial state. A *joint* grid representation (top) utilizes both axes of the grid module 2D space for both task variables, position and accumulated evidence, yielding a wiggling activation pattern (red arrows). A *disjoint* grid representation (bottom) encodes task variables in separate modules, such that each grid module only fires along one axis (red arrows). The periodicity ($\lambda_i$) of each module $i$ is indicated by dashed gray arrows, as the representation space of a grid module is effectively a 2D torus. **(D)** Schematic of the RL setup in which an agent navigates a virtual T-maze with towers appearing on both sides, and a reward is given when it eventually turns to the side with more towers. The agent has some field of view ahead, and this visual sensory input is communicated to HPC through MEC and/or non-grid EC. The HPC code is then used by an RNN policy (cortex) to select an action. The action updates the agent position, which updates the sensory input and grid states. This process repeats throughout the task. **(E)** Model variants that test counterfactual of grid and place cell code, in correspondence to Tables 1, 2.

We leverage the architecture proposed by Chandra et al. (2023), coined *Vector-HaSH+*, as shown in Fig 1B, such that the place cell readouts involve simultaneous projections from both grid cells and non-grid sensory inputs (Fig 1B, orange and green). The grid states are updated by task-relevant velocity input(s) in all or a selective set of modules along axes by design, elaborated in Fig 1C and Appendix A.1. A multilayer perceptron (MLP) is trained to extract evidence velocity from sensory inputs, as shown in Fig 1B (yellow). The updated place cell code is then used as a readout for the RNN policy to make a decision, in which the policy is trained through RL. Although MEC, HPC, and EC can all interact with the cortex biologically (Preston & Eichenbaum, 2013; Eichenbaum, 2017; Canto et al., 2008), we model the place cell vector as the ultimate readout to the cortex given it is minimally sufficient for learning the task and testing the counterfactual of how the co-tuning of place cells arises. This framework enables extensive future studies. For example, one can systematically evaluate the computational advantages of different combinations of {MEC, HPC, EC} input(s) to the cortex for their roles in enabling generalization and rapid learning, *e.g*, EC's sensory inputs may be especially important in scenarios where animals need to remember decision positions in the dark.

## 3.2 MODEL SETUP

Here we describe the model more formally in the context of the accumulating tower task. As the agent navigates in space in time $t$, it processes sensory information from the left and right visual fields, $\vec{f}_L$ and $\vec{f}_R$, which is projected through *i.e.*, the dorsal visual stream to downstream processing regions. This results in a sensory vector in EC modeled by $\vec{s}(t) = \mathbf{W}_R \cdot \vec{f}_L(t) + \mathbf{W}_L \cdot \vec{f}_R(t)$, representing a weighted integration of the two fields that can be used for further computations. Here

we assume a simple concatenation of $\vec{f}_L(t)$ and $\vec{f}_R(t)$, but this technical setup allows flexibility in modeling ablation studies, *e.g.*, experiments that optogenetically inhibit one of the two hemispheres. The downstream computation includes spatial processing and velocity prediction that updates grid cell states and projection into HPC.

In correspondence to Chandra et al. (2023), the MEC layer of the model contains $M$ one-hot grid cell modules, each is a binary-valued periodic function on a 2D discretized hexagonal lattice space with periodicity $\lambda$ (thus each module state is a vector of dimension $\lambda \times \lambda$). The module states are concatenated to form a collective grid state $\vec{g} \in \{0, 1\}^{N_g}$, where the vector length $N_g = \sum_M \lambda_M^2$.

The grid cell state is updated through continuous attractor dynamics, where a module-wise winner-take-all mechanism, $CAN[\cdot]$, shifts each grid module based on position and evidence velocity signals informed by $\vec{s}(t)$, *i.e.*, some form of cohesive velocity estimation through visual-vestibular integration (DeAngelis & Angelaki, 2012), which we model using an MLP (Rosenblatt, 1958). The grid cell state update at time $t$ is thus formalized as

$$\vec{g}(t+1) = CAN[\vec{g}(t)]. \tag{1}$$

The grid cell layer and the non-grid sensory layer project onto the HPC layer, such that the place cell activities are

$$\vec{p}_{\text{mix}}(t+1) = \text{ReLU}[\mathbf{W}_{ps} \cdot \vec{s}(t) + \mathbf{W}_{pg} \cdot \vec{g}(t+1)]. \tag{2}$$

We also test the variants of place cell coding, in which only the grid cell layer projects onto the HPC layer, such that the place cell activities are

$$\vec{p}_{\text{nonmix}}(t+1) = \text{ReLU}[\mathbf{W}_{pg} \cdot \vec{g}(t+1)]. \tag{3}$$

The connectivity between the HPC layer and the EC layer is updated in both cases as pseudo-inverse learned heteroassociative weights,

$$\mathbf{W}_{ps} = PS^+, \tag{4}$$

$$\mathbf{W}_{sp} = SP^+, \tag{5}$$

where $P$ is a $N_p \times N_{patts}$ matrix with $N_{patts}$ hippocampal states, each of length $N_p$, and $S$ is a $N_s \times N_{patts}$ matrix with columns as the encoded sensory inputs of length $N_s$.

The hippocampal state $\vec{p}$ in Eqn 2 (or Eqn 3) serves a readout of the entorhinal-hippocampal circuit to the cortex (under the modeling rationale explained in Section 3.1), which is an action-selection RNN policy trained through policy gradient under reinforcement learning. Please refer to Appendix A.3 for what one step by the agent in the environment entails among the involved brain regions.

## 4 ALTERNATIVE HYPOTHESES OF MULTI-REGION INTERACTIONS

Nieh et al. (2021) observe the existence of conjoint cognitive maps encoding both evidence (cognitive variable) and position (physical variable) in the hippocampus when mice perform the accumulating tower task, suggesting that the hippocampus also performs a general computation, rather than just responding to external stimulus features like space (O'Keefe & Burgess, 1996). This discovery calls for investigation at an appropriate mechanistic modeling scale on how multiple regions coordinate to give rise to internally generated cognition in the brain that gives individuals the ability to flexibly navigate spaces, as well as organize and interrelate experience, objects, and events.

Using the accumulating tower task as a simplistic tool for deriving hypotheses, there are three possibilities that give rise to a conjoint map of physical and cognitive variables in HPC:

- Grid cells encode position, which follows the common belief (Moser et al., 2008), and the conjoint encoding in HPC is due to receiving sensory information from non-grid EC (M2).
- Grid cells co-tune both position and evidence, which is not substantially investigated experimentally to our best knowledge; the EC pathway is not necessary or relevant (M3).
- Grid cells co-tune both position and evidence, and the EC pathway also plays a role in giving arise to the conjoint cognitive map in HPC (M4, M5).

It is worth noting that there are two possibilities for grid cell co-tuning:

Table 1: Overview of how model variants map to testing alternating hypotheses of grid and place cell code that give arise to co-tuning of position and evidence in HPC. Our proposed model, M5, is marked with ∗.

| Models of hypotheses | Evidence is not from MEC | Evidence is from MEC |
|---|---|---|
| Evidence is not from EC | M1 | M3 |
| Evidence is from EC | M2 | M4, M5* |

Table 2: Model variants tested with alternating hypotheses of grid and place cell code that give arise to co-tuning of position and evidence in HPC. Our proposed model, M5, is marked with ∗.

| Model | Grid cell code | Place cell code | MLP input | RNN input |
|---|---|---|---|---|
| M0 | - | - | - | s |
| M1 | pos. | g | - | p |
| M2 | pos. | g + s | - | p |
| M3 | joint pos. & evi. | g | s | p |
| M4 | disjoint pos. & evi. | g + s | s | p |
| M5* | joint pos. & evi. | g + s | s | p |

- **Disjoint integration model**: Individual grid cell modules each encode some task variable, *i.e.*, some grid cell modules exclusively encode position, while others exclusively encode evidence (M4).
- **Joint integration model**: Grid cell modules each encode a combination of evidence and position by leveraging its 2-dimensional toroidal attractor network. This allows for the simultaneous representation of spatial and cognitive variables in grid modules (M5).

Our framework enables a systematic approach to test the above hypotheses, as shown in Table 1 and further detailed in Table 2 by evaluating (a) quantitative learning performance and behavior, (b) qualitative alignment with experimental discoveries, *e.g.*, whether the tested neural mechanism gives arise to choice-specific place cell firing shown in Nieh et al. (2021). We can further hypothesize the potential roles of individual regions in a spatially-embedded decision-making task by accessing if the regions exhibit any structured low-dimensional representation.

## 5 RESULTS

Here we show that our multi-region brain model (M5) governed by biologically plausible neural computation principles yields the prediction of MEC **jointly** integrates position and evidence in a spatially embedded decision-making task represented by the accumulating tower task. Further, we found computational advantages for HPC in leveraging sensory information in the task. Not only does M5 promote rapid learning, but it also aligns with the experimental observations of place cell firing patterns observed in Nieh et al. (2021). Notably, this is the only variant that exhibits visually separable low-dimensional representation of task variables in the Principle Component (PC) space of both HPC and cortical representations, which provides an opportunity for further hypothesizing the role of individual regions in a spatially embedded decision-making task. Please refer to Appendix A.2 for more details, including environment and reward design.

### 5.1 JOINT INTEGRATION OF POSITION AND EVIDENCE IN MEC INDUCES RAPID LEARNING

We present the performance of different model variants in terms of cumulative success rate (Fig 2A) and exploration efficiency (Fig 2B) during training. We found that the agent fails to solve the decision-making task when grid cells do not encode evidence information (Fig 2A, orange and green), implying the MEC is essential in integrating non-innate cognitive variables. Furthermore, we found that jointly tuned grid cells (Fig 2A, red and brown) induce rapid learning, compared to disjointly tuned grid cells (Fig 2A, purple). This is a curious phenomenon without an immediately obvious computational reason, which we further analyze in Section 5.2.

Notably, projecting sensory information onto HPC in M4, M5 leads to unstable learning performance as shown by a higher standard deviation than M3 (Fig 2A), potentially due to the complication of

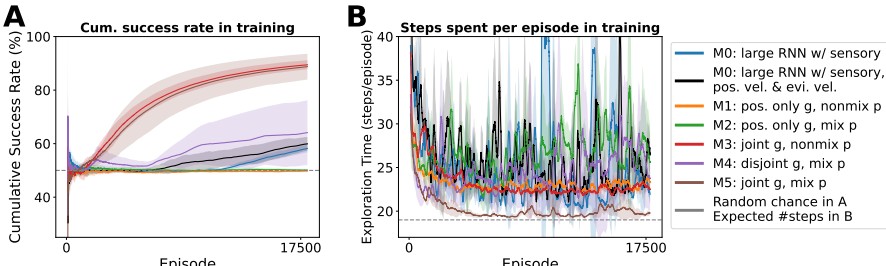

Figure 2: **Learning performance measured by cumulative success rate and exploration efficiency over the course of training for all model variants.** We plot the mean and standard deviation of the metrics across three trials. The baselines are marked using dashed gray lines. In (A), we plot using a window size of 10000 episodes and observe more rapid learning in models with jointly tuned grid cells (M3, in red; M5, in brown). In (B), we plot using a window size of 200 episodes, and we observe M5 efficiently leverages spatial information to navigate through the maze quickly (brown).

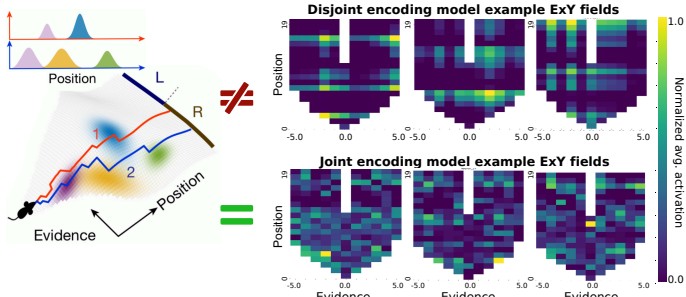

Figure 3: **Place cell tuning during the task.** The schematic plot (left, adapted from Nieh et al. (2021)) and place cell behaviors of selective neurons in M4 (top) and M5 (bottom). The firing field of each neuron is computed by averaging neural activities across trials and normalizing after. Since hippocampal neurons have conjoint evidence and position, smaller firing fields would partition the evidence dimension in E × Y space (left). Consequently, two right-choice trials could produce distinct neural sequences due to different evidence values traversed each trial (left, bottom). Here only M5 with jointly tuned grid cells exhibits expected place fields in E × Y space (right, bottom).

binding mixed place codes with sensory information. Encoding sensory information in HPC, however, increases exploration efficiency (Fig 2B, brown), potentially because the sensory information captures the nuances in the environment, such as where the wall is, supplementary to the rigid information encoded by grid cells. This is supported by M3, a counterfactual of M5 without sensory projection, takes longer to navigate (Fig 2B, red). Please refer to Appendix A.1 for implementation details.

### 5.2 JOINT INTEGRATION MODEL PREDICTS EVIDENCE-POSITION CO-TUNING IN GRID AND PLACE CELLS

According to the quantitative learning metrics present in Section 5.1, the joint tuning of position and evidence in MEC induces rapid learning, while the activation of EC-HPC pathway helps with efficient spatial navigation, among simulated hypotheses. Here we present a more in-depth analysis of the relationship between the grid cell computation and place cell firing patterns. We show that the model variants that rapidly learn the task exhibit firing fields that most closely mimic the experimental observations made in Nieh et al. (2021), yielding an immediate prediction of the underlying grid cell computation that supports the joint encoding of spatial and cognitive variables in hippocampus.

#### 5.2.1 ONLY JOINT INTEGRATION MODEL GIVES RISE TO EXPERIMENTALLY ALIGNED JOINT HPC FIRING FIELDS

Nieh et al. (2021) confirmed experimentally that individual CA1 neurons encode both position and accumulated evidence. Due to this interdependence of position and evidence, one should expect trials of the same final decisions would evoke different hippocampal neuronal firing sequences due to different configurations of evidence traversed by the agent (Fig 3 left). Thus, smaller firing fields would divide the evidence dimension in Evidence (E) × Position (Y) space of hippocampal activities.

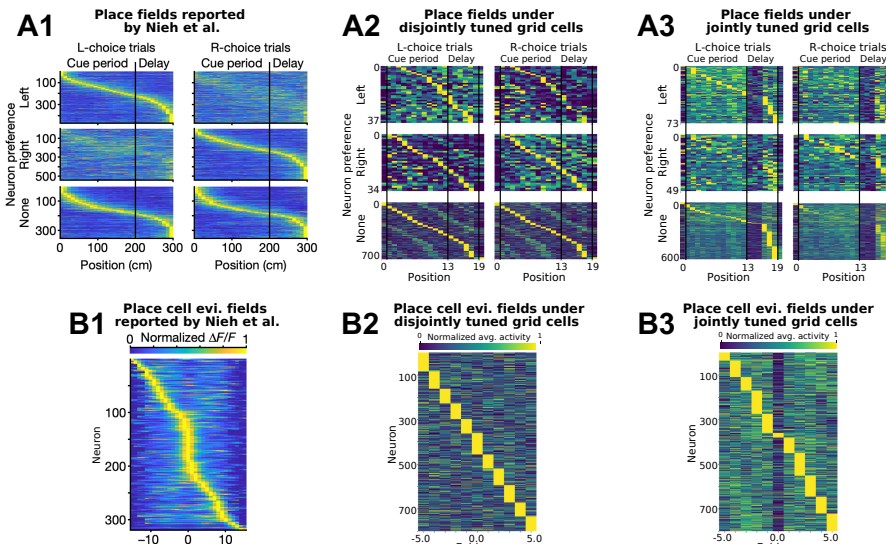

Figure 4: **Choice-specific place cell sequences and evidence fields.** We show that while place cells form evidence fields when grid cells encode evidence, only joint integration models exhibit choice-specific place cell sequences when activities are sorted by the positions of peak activities, aligning with the experimental results in Nieh et al. (2021). We postulate the noise in firing fields could be attributed to the encoding of both evidence and position, similarly indicated in Nieh et al. (2021) where they observe unreliable response of individual cells on a trial-by-trial basis compared to a baseline alternation task. **(A)** Choice-specific place cell sequences in Nieh et al. (2021) (A1), disjoint integration model M4 (A2), and joint integration model M5 (A3), divided into left-choice-preferring (top row), right-choice-preferring (middle row), and non-preferring (bottom row) cells by evaluating the significance of mutual information. Cells are shown in the same order within each row group, sorted by peak activities. The average activation was normalized within each neuron. **(B)** The firing fields of place cells in accumulated evidence space appear when sorted by the positions of peak activities, in Nieh et al. (2021) (B1), disjoint integration model M4 (B2), and joint integration model M5 (B3). The average activation was normalized within each neuron.

Additionally, since the cognitive maps would compromise at least two dimensions, including position and evidence, the firing fields evaluated in a single dimension would exist but could appear unreliable across trials of varying configurations.

In Fig 3 right, we demonstrate that the joint integration models (M5 in Fig 3 right, bottom; M3 in Appendix C) successfully replicate the $E \times Y$ place fields indicated by Nieh et al. (2021). In contrast, model variants that do not incorporate joint integration, such as the disjoint integration model (Fig 3 right, top), fail to reproduce this behavior. These models primarily rely on independent representations of position or evidence as shown by the stripe firing patterns.

## 5.3 JOINT INTEGRATION MODEL GIVES RISE TO BOTH EVIDENCE FIELDS AND CHOICE-SPECIFIC PLACE FIELDS

Here we show that only the joint integration model perfectly aligns with the experimental results, strongly validates the joint integration model governs the neural computational rules that give rise to place cell behaviors established in Nieh et al. (2021), in which grid cells jointly encode evidence and position such that hippocampus creates conjoint maps.

**Only joint integration models exhibit choice-specific neurons** Nieh et al. (2021) show that CA1 neurons exhibited choice-specific place cell sequences when sorted by the positions of peak activities. To ensure a fair comparison, we employ the same method to analyze our models. Specifically, we measure the mutual information (see Appendix B) between the neural activity of each cell and the agent's position during left- and right-choice trials, and compare it to a shuffled dataset. Our results show that, exclusively in joint integration models (M3, M5), a small fraction of place cells are

choice-specific under this metric. Additionally, place cells exhibit choice-specific place fields when sorted by their peak activity positions (Fig 4, A3; Appendix D, B), consistent with experimental observations (Fig 4, A1). In contrast, the disjoint integration model fails to produce choice-specific place cell sequences (Fig 4, A2).

**Place cells formed firing fields in evidence space**    Similarly, we measure the mutual information between accumulated evidence and the neural activity of each place cell. As expected, when grid cells encode evidence, place cells formed firing fields in evidence space that spanned small segments of evidence values (sorted by the position of peak activity), consistent with Fig 3, left. However, place cells do not form firing fields in evidence space in M1 or M2, *i.e.*, when evidence information is absent or from EC rather than being encoded in grid cells (see Appendix D).

### 5.3.1 ONLY JOINT INTEGRATION MODEL WITH ACTIVATED EC PATHWAY EXHIBITS WELL-SEPARATED LOW-DIMENSIONAL CO-REPRESENTATION OF TASK VARIABLES

We perform Principal Component Analysis (PCA) on hippocampal and cortical activities to identify visually separable clusters in PC space when colored by task variables. The presence of low-dimensional representations would provide insight into the functional roles of specific brain regions and the utility of different neural computational rules tested by model variants.

We found the joint integration model with activated EC pathway (M5) uniquely exhibit visually separable clusters of hippocampal activity in PC space for both position (Fig 5, B1) and local evidence velocity (#R-#L tower at a position, Fig 5, B2), in contrast to all other variants (Figs 5, A1, A2, and Appendix E). We did not observe separability in accumulated evidence in hippocampal activities in first 3 PCs: this is counter-intuitive given grid cells encode and communicate accumulated evidence to hippocampus, while EC does not encode fine-grained information. Given hippocampal neurons are projection neurons (Fox & Ranck Jr, 1981; 1975), it is curious what the source and encoding mechanism of local evidence velocity is in hippocampus. We suggest future studies to analyze existing experimental data to validate the predicted hippocampal role of representing local evidence velocity, and to further dissect the proposed model through ablation studies for understanding the underlying computational mechanism. Overall, this suggests that the integration of sensory information from the EC is necessary for creating a cohesive, low-dimensional representation of relevant task variables in hippocampus, supporting rapid learning and efficient spatial navigation (Fig 2).

Additionally, we observe the separability of three possible agent actions in hippocampal and cortical activities in M4 and M5 (Fig 5, panel 3 and 4). The separability of actions exists in M1 and M2 (Appendix E), which are models that cannot learn the accumulating tower tasks but are limited to clusters of going forward and clusters of one particular turning action. The separation of actions is noisy in M3 when sensory information from EC is absent (Appendix E). These observations suggest hippocampal cognitive maps, emerged from jointly integrated grid cells and sensory information, are self-contained and sufficient for downstream decisions.

Together, this stream of results driven by hypotheses tested in mechanistic models reveals necessary neural mechanisms for forming cohesive and low-dimensional representations of task variables in hippocampal cognitive maps. In particular, we predict and demonstrate conjoint tuning in grid cells and sensory pathways are critical for mimicking the biology observed in experiments, and giving rise to important behaviors such as rapid learning and efficient spatial navigation (Fig 2).

## 6 DISCUSSION

Our work highlights the critical role of conjunctive encoding in grid cells, where both spatial and task-relevant information are jointly represented. We propose a multi-region brain model combining entorhinal-hipocampal circuit (Chandra et al., 2023) and RNN which works as neocortex and evaluate it in accumulating tower task (Nieh et al., 2021), which we formulate as reinforcement learning problem. Our model demonstrates that this integration, along with non-grid sensory inputs from the entorhinal cortex, is crucial for rapid learning and decision-making in spatially embedded tasks. These findings align with experimental observations, showing that the hippocampus forms cognitive maps that combine spatial locations and accumulated evidence, enabling efficient navigation and flexible decision-making.

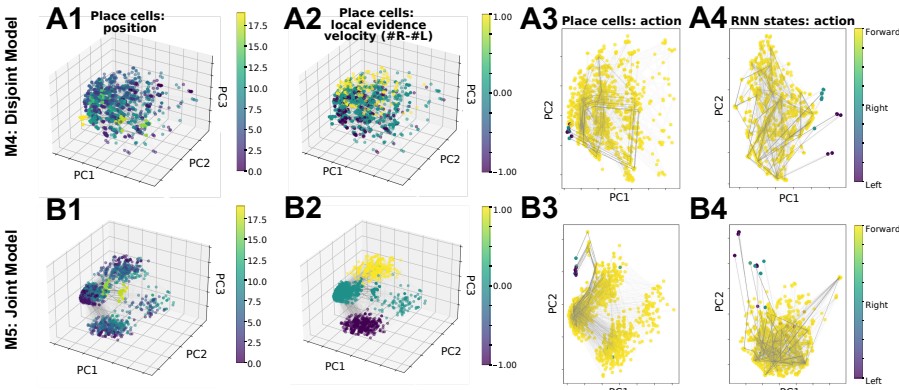

Figure 5: **Separability of hippocampal (column 1, 2) and cortical representations (column 3, 4) in low-dimensional PC space for M4 (row A) and M5 (row B).** Joint grid cell integration with activated non-grid EC-HPC pathway (M5) uniquely leads to separable neural activity clusters in hippocampus when colored by position (B1) and local evidence velocity (#R-#L per position, B2). We did not observe other separable task variables in the first 3 PCs of hippocampal activities in all variants (Appendix E). We observe the separability of actions in hippocampal and cortical activities in PC space in both M4 and M5 (column 3, 4).

The prediction of conjunctive encoding of spatial and task-relevant information in grid cells reinforces the hypothesis that spatial and cognitive processes are deeply intertwined in the brain's navigation and memory systems (Buzsáki & Moser, 2013). Neural algorithms that support path integration and navigation are likely repurposed for more abstract forms of cognition, enabling the hippocampal-entorhinal network to facilitate both physical navigation and decision-making based on internal cognitive states. However, sensory inputs to the hippocampus, while enhancing spatial efficiency, introduce variability in learning performance, suggesting a trade-off between precision and complexity in neural representations. Future research should investigate how different neural inputs optimize decision-making processes in diverse environments.

While our proposed framework did not consider the role of CA3 recurrence, we observe that adding this mechanism does *not* induce the experimentally observed place cell phenomena (Nieh et al., 2021) in M2 and M4, as demonstrated in Appendix F. It is worth noting that the consideration of CA3 recurrence is complementary to our predicted grid cell coding scheme, and these mechanisms are not mutually exclusive. Our framework can be extended to explore the causal effects of CA3 recurrence more thoroughly, through ablation studies incorporating models similar to M1 through M5 but with CA3 recurrence included and other mechanisms selectively excluded. Furthermore, we are actively collecting neurophysiological data to test our falsifiable prediction. The results of these experiments will directly inform whether additional mechanisms, such as CA3 recurrence, contribute to place cell tuning, and can be readily investigated within our current framework.

It is also worth noting that the use of an MLP to extract velocity signals is a simplification of visual-vestibular integration (DeAngelis & Angelaki, 2012), justified as a consequential assumption from experimentally well-supprted (Malone et al., 2024; Yoon et al., 2013; Alme et al., 2014) Vector-HaSH (Chandra et al., 2023), that the velocity signals are processed by a potentially diverse set of brain regions external to the entorhinal-hippocampal circuit (Morris & Derdikman, 2023), and is then fed into grid cells. This simplification of the underlying velocity abstraction process does not affect our results, and is not the focus or within the scope of our framework.

In conclusion, our findings provide a comprehensive framework for understanding how the hippocampal-entorhinal-neocortical network integrates physical and cognitive information to create flexible cognitive maps that support learning and navigation. Our work opens pathways for future experimental studies to explore these straightforward, falsifiable predictions and understand how the brain generalizes mechanisms of physical navigation to solve abstract cognitive problems. Moreover, this serves as a strong example of how machine learning can be applied to neuroscience and vice versa to explain neural phenomenon, guide future research directions, and demonstrate the efficiency of neuro-inspired artificial intelligence.

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

# APPENDIX

## A EXPERIMENTAL DETAILS

### A.1 MODEL

For M1-M5, we use three grid modules, with corresponding periodicity of $7, 8, 11$, thus $N_g = 234 \, (= 7^2 + 8^2 + 11^2)$. We simulate $800$ place cells. The MLP and RNN both have a learning rate of $0.0005$ and a hidden size of 32. The RNN has leaky units, with $\alpha = 0.025$.

The grid coding scheme is by design, as illustrated by an ealier example in Fig 1C. Generally, the velocity inputs to each grid module updates the grid phases through path integration, following Vector-HaSH implementation in Chandra et al. (2023). The evidence velocity to grid cell modules is effectively the number of towers on the right minus the number of towers on the left in the current position, which is predicted through an MLP from the current field of view (sensory inputs). The positional velocity, on the other hand, is represented as 0 (stuck) or +1 (forward) without the use of MLP, given backward-moving is not task-relevant (Nieh et al. (2021), behavioral training), though it is possible to use an MLP as well. We made a simplification here without the use of MLP, given this change will not affect the results.

For the standalone RNN baselines (M0), we scale up the hidden size to $32 + N_g + N_p + N_s = 1076$ to ensure the total number of neurons used is consistent with M1-M5. The input to RNN baselines is sensory information, with one of the variants additionally takes in positional velocity (moved or not) and evidence velocity (predicted by MLP with the same setup as M1-M5). We use a learning rate of $0.0001$ with the gradient clipped to a max norm of 1. This is because the large standalone RNNs fail to learn with the learning rate of $0.0005$ used by M1-M5 due to learning instability (*e.g.*, exploding or vanishing gradients). As a result, we conducted a hyperparameter search and used the settings that yielded the best performance.

### A.2 ENVIRONMENT

The accumulating-tower task involves an agent moving through a T-maze with towers on both sides (Fig 1C) and must turn at the end toward the side with more towers to obtain a reward. We divide the maze into non-overlapping start region (9%, no towers), cue region (61%, with towers), and delay and decision region (the rest, no towers), in approximate alignment with Nieh et al. (2021). We represent each episode as a unique configuration of towers, involving several steps down the corridor until an agent reaches the T-arm and makes a turn. We represent the left and right sides of the maze each as a vector ('1' = tower, '0' = non-tower, '−1' = outside-of-maze). The agent has a field of view on some number of positions ahead.

The 'rewarded' side (with more towers) in each episode is chosen uniformly at random, with $x_{\text{reward}}$ number of towers. Specifically, the number of towers on the reward side $x_{\text{reward}}$ is sampled from $\text{Uniform}(1, K)$ with $K$ being some maximum threshold. The 'non-rewarded' side has strictly fewer towers $x_{\text{non-reward}}$, where $x_{\text{non-reward}} \sim \text{Uniform}(0, x_{\text{reward}})$. At each step, the agent can move $\texttt{left}(0)$, $\texttt{right}(1)$, or $\texttt{forward}(2)$. Before reaching the end, the agent is rewarded each step $0.01$ for going forward and a penalization of $−0.001$ otherwise. After reaching the end, the agent gets a reward of $10$ for a correct turn, no reward for the opposite turn, and $−1$ for continuing forward (bumping the wall) every time; the episode ends when the agent makes a turn or reaches a maximum number of attempts (which is penalized by $−5$). We simulate a maze sequence length of $20$ (so the division of the start, cue, delay and decision region length is $\{1, 12, 6, 1\}$ correspondingly), a field of view of $5$, and train the RNN using REINFORCE algorithm (Sutton et al., 1999) until convergence.

## A.3 SINGLE STEP IN THE TASK

One step in the accumulating task involves the following (Fig 1C): upon the agent perceives the sensory information, an MLP (is trained to) abstracts evidence velocity from the processed sensory information perceived within the field of view. The grid cell state is consequently updated by position and evidence velocity signals, and projected onto place cells along with the non-grid sensory vector. The place cell code is fed to the cortical RNN policy to select an action on going left, right, or forward. The succeeding position update (stay or advance a position) carries relevant sensory information, which is again fed implicitly to the grid cell subnetwork to shift the grid states; the grid vector and the corresponding newly perceived sensory information are used to update place cell states again. This process repeats until the agent eventually reaches the T-arm and makes a turning decision while following the reward scheme mentioned in Appendix A.2.

# B MUTUAL INFORMATION

## B.1 MUTUAL INFORMATION ANALYSIS

We follow the mutual information analysis in Nieh et al. (2021). Here we reiterate this procedure for completeness. For each neuron, we evaluate the mutual information metric defined in Skaggs et al. (1992),

$$I = \int_x \lambda(x) \log_2 \frac{\lambda(x)}{\lambda} p(x) dx,$$

in which $I$ is the mutual information rate of the neuron in bits per section, $x$ is the spatial location (or accumulated evidence) of the agent, $\lambda(x)$ is the mean firing rate of the neuron at location (accumulated evidence) $x$, $p(x)$ is the probability density of the agent occupying location (accumulated evidence) $x$ and $\lambda = \int_x \lambda(x) p(x) dx$ is the overall mean firing activity of the neuron.

## B.2 SCATTERPLOTS OF MUTUAL INFORMATION

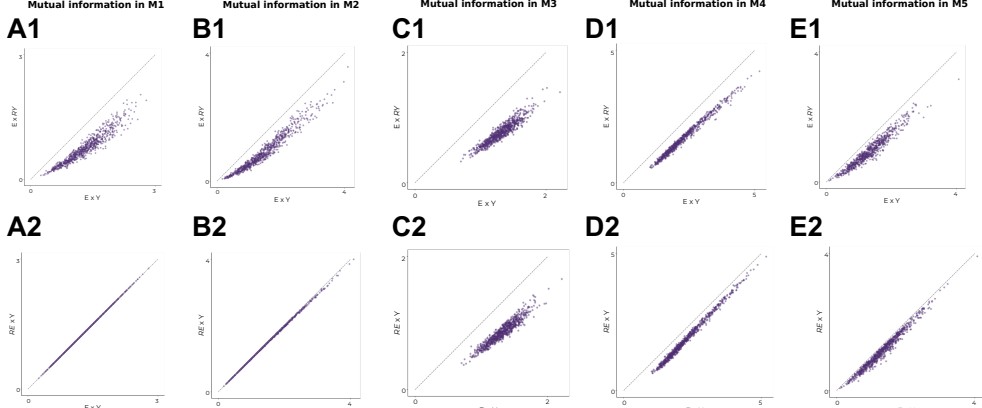

Figure 6: Scatterplots of the hippocampal mutual information in $E \times RY$ space versus $E \times Y$ space (top row), and scatterplots of mutual information in $RE \times Y$ space versus $E \times Y$ space (bottom row). We show data for all model variants M1 to M5, in the order of panel A to $E$ respectively. We observe evidence and position interact to provide meaningful information in M3, M4, and M5 (when grid cells co-tune position and evidence), while M1 and M2 rely on information of position (when grid cells tune evidence only, and there is either no or some sensory information projected into hippocampus). Here, $RY$ is a randomized position, generated by randomly sampling from the $Y$ distribution that corresponded to the non-randomized E value of the cell. A similar procedure is performed for generating the $RE \times Y$ variables. More details of the procedure are described in the Mutual Information Analysis section of Nieh et al. (2021).

## C   HIPPOCAMPAL FIRING FIELDS WITHIN $E \times Y$ SPACE IN MODEL VARIANTS

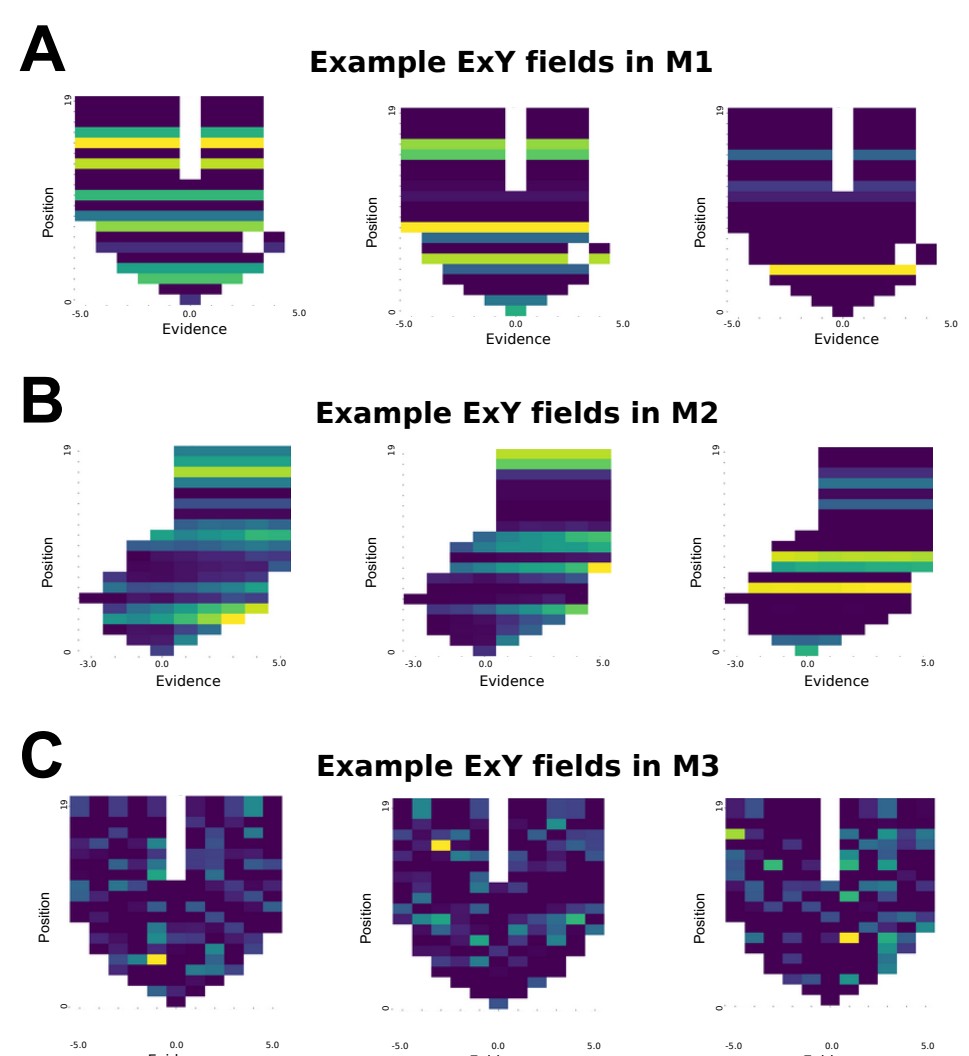

Figure 7: Example hippocampal firing fields in $E \times Y$ space in M1 (A), M2 (B), and M3 (C). We observe the firing fields of M1 and M2 (A,B) do not depend on evidence with stripe patterns. M2 firing fields occasionally have some amount of gradient, a potential artifact of sensory injection, similar to the firing fields of M4 and M5 in Fig 3. M3 (C) firing fields exhibit conjoint tuning of position and evidence and have no apparent gradient artifacts.

# D  HIPPOCAMPAL EVIDENCE AND PLACE FIELDS IN MODEL VARIANTS

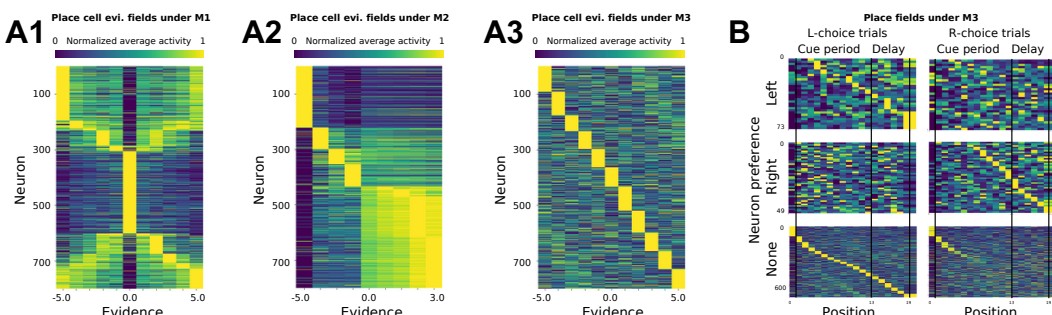

Figure 8: Hippocampal firing fields in evidence (A) and in space (B), for M1 (A1), M2 (A2), and M3 (A3, B). We see M1 and M2 do not have firing fields in evidence (A1, A2), while M3 does (A3). Furthermore, M3 contains choice-specific place fields (B) similar to M4 (Fig 4, A3), implying joint tuning of position and evidence in grid cells is key to forming conjoint hippocampal map.

# E HPC AND RNN PC REPRESENTATION IN MODEL VARIANTS

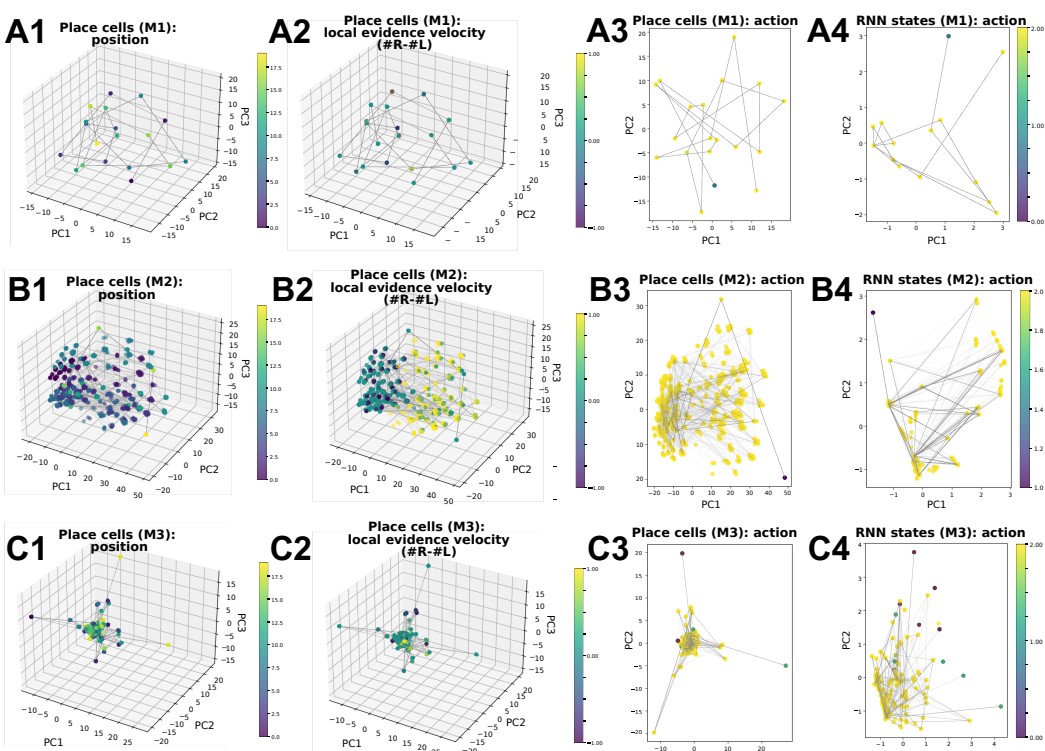

Figure 9: Low-dimensional representation of hippocampal and RNN activities in PC space, shown for M1 (row A), M2 (row B), M3 (row C). We show the representations colored according to selective task variables, specifically position, local evidence velocity, and action, in which M4 shows clear separation (in correspondence to Fig 5). Other variables visualized in HPC and RNN activity PC space include accumulated evidence, position changes, left-/right-choice trials, total evidence of the trial, and ground truth action; we observe no visual separation.

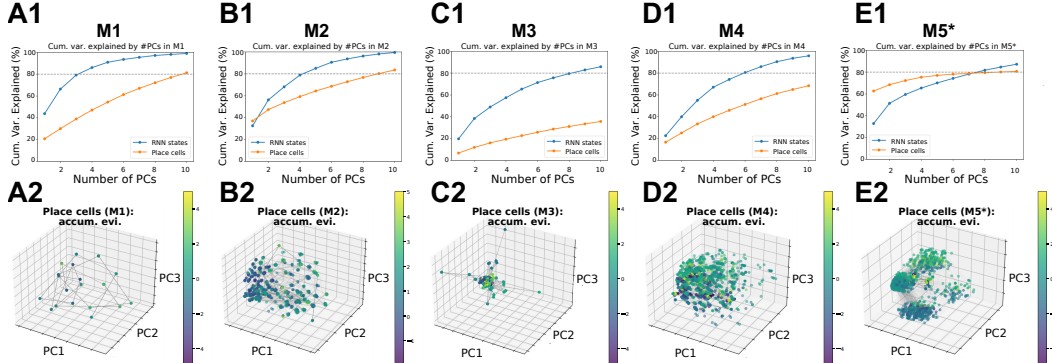

Figure 10: Cumulative variance explained in percentage of hippocampal (orange) and RNN (blue) activities, by number of principle components (PCs), and low-dimensional representations of hippocampal activities in PC space, colored by accumulated evidence, shown for M1 (column A), M2 (column B), M3 (column C), M4 (column D), and M5 (column E). The first two PCs in M5 explained the most amount of variance in hippocampal representations (68%) in comparision to other model variants. We do not observe any visual separability of accumulated evidence in PC space of the first three PCs, as shown in the second row.

## F    EFFECT OF CA3 RECURRENCE IN M2 & M4

In this section, we demonstrate in M2 and M4, as a proof of concept, that the inclusion of CA3 recurrent connectivity in the HPC layer does not affect the general conclusions presented in the main paper. Specifically, the inclusion of CA3 recurrence in M2 or M4 does not induce the experimentally observed place cell phenomena (Nieh et al., 2021), producing similar results as if recurrence was absent (see Figs 3, 4, 6, 7, and 8).

To model CA3 recurrence, we incorporate additional recurrent connections within the HPC layer, $\mathbf{W}_{pp}$, updated through associative learning using $\vec{p_{\text{mix}}}(t)$ and $\vec{p}\text{mix}(t+1)$, analogous to the learning process for $\mathbf{W}_{ps}$ and $\mathbf{W}_{sp}$ (see Eqns. 4 and 5). The activity of mixed place cells is then described by:

$$\vec{p}_{\text{mix}}(t+1) = \text{ReLU}[\mathbf{W}ps \cdot \vec{s}(t) + \mathbf{W}_{pg} \cdot \vec{g}(t+1) + \mathbf{W}_{pp} \cdot \vec{p_{\text{mix}}}(t)]. \tag{6}$$

The rest of the setup remains consistent with Appendix A.1.

As shown in Fig. 11, the inclusion of recurrent integration of positional information from MEC and sensory information from non-grid EC does not result in the emergence of conjunctive place cells in M2. Similarly, Fig. 12 demonstrates that the lack of conjunctive place cells in M4 persists when grid cell modules encode position and evidence disjointly.

These findings confirm that the recurrent integration in HPC alone does not induce conjunctive coding, underscoring the critical role of joint integration of position and evidence in grid cells for producing co-tuned place cells.

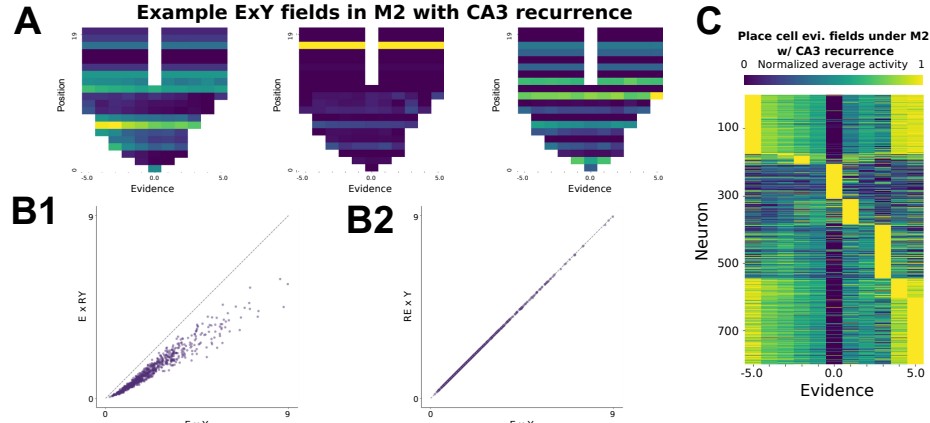

Figure 11: **Analysis of hippocampal code in M2 with CA3 recurrence. (A)** Example hippocampal firing fields in $E \times Y$ space. **(B)** Scatterplots of the hippocampal mutual information in M2 with CA3 recurrence when only the position is randomized (B1), and when only the evidence is randomized. The model shows higher mutual information in position only. See the caption of Fig B.2 for implementation details. **(C)** Hippocampal firing fields in evidence in M2 with CA3 recurrence.

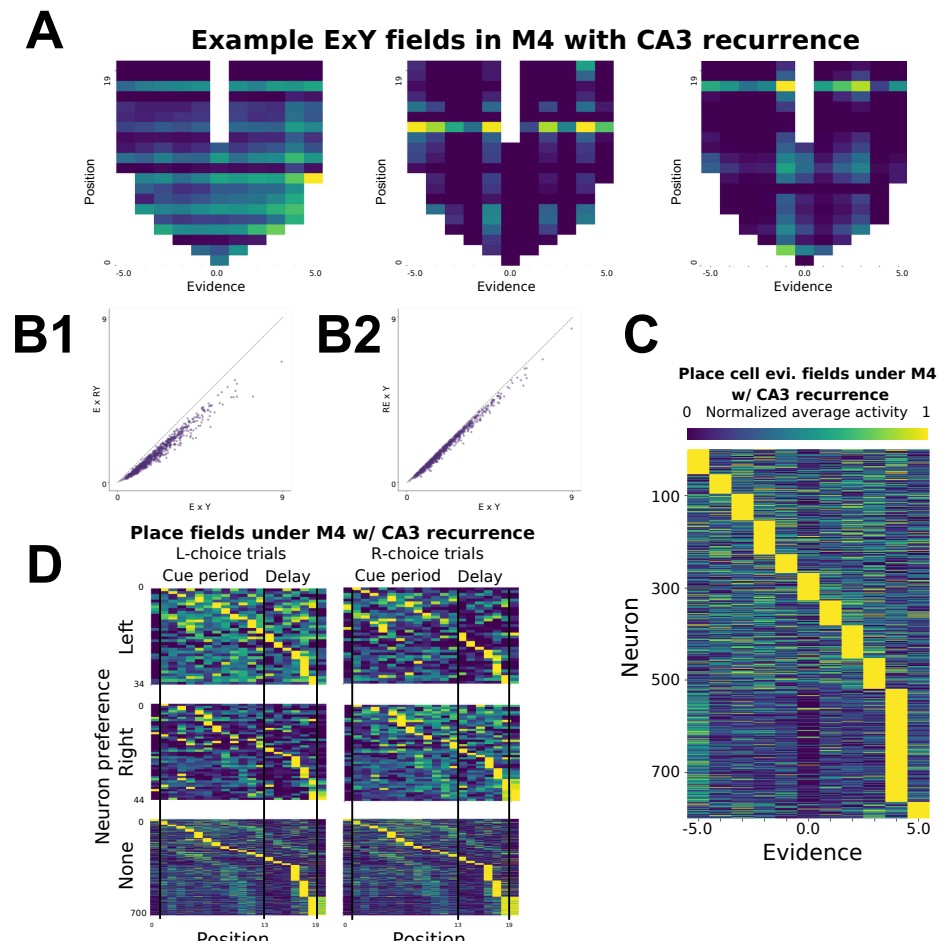

Figure 12: **Analysis of hippocampal code in M4 with CA3 recurrence.(A)** Example hippocampal firing fields in $E \times Y$ space. **(B)** Scatterplots of the hippocampal mutual information in M4 with CA3 recurrence when only the position is randomized (B1), and when only the evidence is randomized. The model shows higher mutual information in both position and evidence, consistent with the case when the recurrence is not considered (Fig B.2, column D). See the caption of Fig B.2 for implementation details. **(C)** Hippocampal firing fields in evidence in M4 with CA3 recurrence. We still observe evidence fields. **(D)** Hippocampal firing fields in space. We do not observe choice-specific place fields shown in Nieh et al. (2021) after considering the CA3 recurrence in M4.