# OpenReview forum: "A multi-region brain model to elucidate the role of hippocampus in spatially embedded decision tasks"
_ICLR.cc/2025/Conference — Submitted to ICLR 2025_

### Official Review · Reviewer_qnRf · 2024-11-01

**Soundness:** 2
**Presentation:** 3
**Contribution:** 2
**Rating:** 5
**Confidence:** 4

**Summary:**

The authors proposed a multi-regional reinforcement learning model to solve the evidence accumulation task. The model comprised of MEC for grid cells, EC for sensory cells, hippocampus for place cells, RNN as neocortex and perhaps the policy network as the striatum. The authors proposed 5 different model variants with different network connectivity and information passed into the hippocampus layer and evaluated its learning behavior and analyzed its representation. Based on the results, the authors propose that place cells that conjunctively integrate position and sensory information was important to solve the task.

**Strengths:**

-- clearly explained the rationale for the model development.
-- presented 5 hypothesis for grid-place interaction for evidence accumulation navigation and showed how they deferred across behavior and representation.
-- The authors demonstrated that the model recapitulates the neural representations observed in experiments.

**Weaknesses:**

-- The authors only demonstrated that their model replicated the representations observed in 1 experiment (Nieh et  al. 2021). The authors should consider recapitulating 1 other neural phenomena to increase the generality of their proposed model e.g. representational drift (Qin et al. 2023 Nature Neuro.), or increase the variety of simulations e.g. 5 arm decision navigation (Baraduc, Duhamel, Wirth, 2019 Science).

-- Although the title specifies the role of hippocampus (instead of the entorhinal cortex), the authors only varied their models using different entorhinal activity to formulate the hypothesis in Table 1. This could be to demonstrate how the type of input to the hippocampus influences navigation learning behavior. Nonetheless, the authors should consider previous models that use different hippocampal representations for navigation learning (Brown & Sharp, 1995; Arleo & Gerstner, 2000; Foster et al. 2000; Zannone et al. 2018; Kumar et al., 2022). For instance, will an RNN based RL agent with place cells and sensory evidence as input learn to solve the task (Kumar et al. 2022 Cerebral Cortex, https://doi.org/10.1093/cercor/bhab456; Singh et al. 2023 Nat Mach. Int., https://doi.org/10.1038/s42256-022-00599-w), similar to M0 but with place cell position inputs? This proposal also suggests a EC-MEC-hippocampus-neocortical-striatum pathway but was not considered.

-- the claim that the model enables rapid learning in spatially embedded task is not warranted. how do the authors define rapid learning and efficient navigation (red and brown curves in Fig. 2)? These agents still require close to 1000 episodes to converge instead of 1 or a few episodes (Foster et al. 2000 Hippocampus https://doi.org/10.1002/(SICI)1098-1063(2000)10:1<1::AID-HIPO1>3.0.CO;2-1 ; Kumar et al. 2024 https://doi.org/10.48550/arXiv.2106.03580)? Rapid learning is usually used under the context of zero, one, or few-shot learning, and not thousands of trials.

-- it is unclear why M3 and M5 show similar increase in success rates of learning (Fig. 2A) when M3 does not get evidence from EC? Additionally, it is unclear why M0 and M5 show similar decrease in steps spent per episode (Fig. 2B). Representational analysis for M0 and M3 should be included in Fig. 3,4 and 5.

**Questions:**

-- the authors proposed a mechanistic hippocampus-entorhinal setup. However, it is unclear whether they used a biologically plausible learning rule or backpropagation for policy learning i.e. policy gradient. The former will make the model very appealing. But this is a minor point.

-- why is there a disparity in M0's and M5's learning performance in Fig. 2A and 2B? Shouldn't we expect a model with a slower increase in cum. success rate to demonstrate a slower decrease in steps spent per episode? Maybe I am missing something.

-- Since separability of clusters is not a prerequisite for navigation behavior (M3 vs M5 Fig. 9 vs Fig. 2), how can we make sense of the representation to behavior?

---

> ### Author Response · Authors · 2024-11-18
>
> *Thread [1/3]*
>
> Thank you very much for your detailed, insightful feedback and the mention of many related valuable works. We highly appreciate your time helping us to improve the quality of our work. Here, we address your feedback and questions in order, in addition to our modification in the manuscript following your feedback, such as in the Discussion and interpretation of results, among others, all in blue.
>
> > Based on the results, the authors propose that place cells that conjunctively integrate position and sensory information was important to solve the task.
>
> This is most likely a typo in the summary. But in case of any misunderstanding, we’d like to clarify that we predicted grid cells (not place cells, which are already experimentally established) that conjunctively integrate position and sensory information, along with an activated non grid EC-HPC pathway, were important to solve the task and replicate experimental findings of conjunctive tuning in place cells.
>
> **Regarding weaknesses:**
>
> > The authors only demonstrated that their model replicated the representations observed in 1 experiment (Nieh et al. 2021). The authors should consider recapitulating 1 other neural phenomena to increase the generality of their proposed model e.g. representational drift (Qin et al. 2023 Nature Neuro.), or increase the variety of simulations e.g. 5 arm decision navigation (Baraduc, Duhamel, Wirth, 2019 Science).
>
> We believe the current scope of our work is well contained. Our work studies multi-region interactions underlying spatial navigation given recent findings indicate the same circuit has a potential role in spatial decision-making (Nieh et al., 2021). The findings of the work is binded to cognitive map theory. Though it is conveniently doable to simulate additional tasks that are not directly relevant, we think adding these tasks would deviate away from the main focus of the paper.
>
> > Nonetheless, the authors should consider previous models that use different hippocampal representations for navigation learning (Brown & Sharp, 1995; Arleo & Gerstner, 2000; Foster et al. 2000; Zannone et al. 2018; Kumar et al., 2022).
>
> Thank you very much for your insights! We would appreciate more clarification on the first proposal, or, if you were referring to a potential inclusion of CA3 recurrence, we have added a new paragraph to Discussion justifying our current choice of modeling (lines 516-525).
>
> > For instance, will an RNN based RL agent with place cells and sensory evidence as input learn to solve the task (Kumar et al. 2022 Cerebral Cortex; Singh et al. 2023 Nat Mach. Int.), similar to M0 but with place cell position inputs? This proposal also suggests a EC-MEC-hippocampus-neocortical-striatum pathway but was not considered.
>
> Regarding the latter, this is a great point, and we actually have results on the variant of M0 that receives both hippocampal position code and sensory, as well as variants to M1-M5 where RNN receives both hippocampal code and sensory. These are not ultimately included because of the scope of the work, as we think these results will bring added confusion without ultimately affecting the current result. Our consideration and justification of only considering p → RNN is discussed in the last paragraph of Section 3.1 lines 202-208. Below we copy the content directly for your convenient reference:
>
> **“** Although MEC, HPC, and EC can all interact with the cortex biologically, we model the place cell vector as the ultimate readout to the cortex given it is minimally sufficient for learning the task and testing the counterfactual of how the co-tuning of place cells arises. This framework enables extensive future studies. For example, one can systematically evaluate the computational advantages of different combinations of {MEC, HPC, EC} input(s) to the cortex for their roles in enabling generalization and rapid learning, e.g, sensory inputs from the EC may be especially important in scenarios where animals need to remember decision positions in the dark. **”**

---

> > ### Author Response · Authors · 2024-11-18
> >
> > *Thread [2/3]*
> >
> > **Continued: regarding weaknesses**
> >
> > > the claim that the model enables rapid learning in spatially embedded task is not warranted. how do the authors define rapid learning and efficient navigation (red and brown curves in Fig. 2)? …Rapid learning is usually used under the context of zero, one, or few-shot learning, and not thousands of trials.
> >
> > Thank you for clarifying our terminology, and pointing us to valuable works in few-shot learning. The goal of the study is indeed not few-shot learning, but instead offering a biologically plausible testbed for exploring hypotheses of neural computation underlying multi-region brain interactions in spatially embedded decision-making tasks, formulated using reinforcement learning. Our use of “rapid learning” is meant to be relative within our model variants (hypotheses) in the context of the paper, since the comparison is not otherwise fair. Though we stated this in most parts of our paper, e.g., in the abstract (line 22), and mostly mentioned the term when comparing model variants, e.g., lines 319-320, we understand this can be confusing. We have modified our wording to “faster”, “more rapid”, or similar terms as appropriate to make it more clear. We’d also like to add that the model behavior reflects the reinforcement learning behavior in reality. In experiments, the mice underwent at least 11 shaping stages of tasks in increasing difficulty. Mice were typically trained 5–7 days/week, for one 1-h session per day, and took 6-7 weeks to learn the accumulating tower task (Pinto et al. https://doi.org/10.3389/fnbeh.2018.00036).
> >
> >
> > > it is unclear why M3 and M5 show similar increase in success rates of learning (Fig. 2A) when M3 does not get evidence from EC?
> >
> > We would like to clarify our methodology. To reiterate, M3 gets a joint grid code of both evidence and position from MEC, the same as M5, as indicated in Table 1 and 2. The difference between M3 and M5 is whether the non-grid EC-HPC pathway is activated; it is not activated in M3, but activated in M5, also indicated in both tables. According to the experimental results shown in Fig 2, 6, 7, 8, the projection of conjunctive grid code to HPC alone is sufficient to give rise to conjunctive place code, which could explain why M3 and M5 have similar increase in success rates of learning.
> >
> > > Additionally, it is unclear why M0 and M5 show similar decrease in steps spent per episode (Fig. 2B).
> >
> > Here's our intuition regarding the navigation efficiency of M0 and M5: non-grid sensory input to HPC (or sensory to RNN) is potentially helpful to capture the nuances in the environment, such as where the wall is (marked as ‘-1’, hinting the decision region). This source of nuances is present in M0, M4, and M5, and we indeed observe fast navigation in both M0 and M5. M4 does not exhibit fast navigation, which can be due to other confounds such as its disjoint grid code. This would similarly explain why M3’s increase in success rate is comparable to M5, but is slower in navigating. For added clarity, we added lines 358-360 in blue. Thank you for your feedback!
> >
> > > Representational analysis for M0 and M3 should be included in Fig. 3,4 and 5.
> >
> > Representation analysis for model variants is already included in the Appendix, such as Figure 7,8,9,10, to not deviate the attention from the comparison of joint vs disjoint grid code. This additional information in the Appendix is referred to throughout the main content when related results are discussed. A representation analysis for M0 would not be comparable to M1-M5 as it does not receive hippocampal input like other variants, therefore it is not valuable to the focus of our study; M0 should be treated as a baseline when an abstractor network, i.e., Vector-HaSH, is absent.

---

> > > ### Author Response · Authors · 2024-11-18
> > >
> > > *Thread [3/3]*
> > >
> > > **Regarding questions:**
> > >
> > > > However, it is unclear whether they used a biologically plausible learning rule or backpropagation for policy learning i.e. policy gradient.
> > >
> > > The connectivity between non-grid EC and HPC is learned through Hebbian learning (Equation 4, 5). We used a policy gradient (REINFORCE) for RNN. The RL training of RNN was briefly described in lines 247-248, “which is an action-selection RNN policy trained through policy gradient under reinforcement learning…”
> > >
> > > > why is there a disparity in M0's and M5's learning performance in Fig. 2A and 2B? Shouldn't we expect a model with a slower increase in cum. success rate to demonstrate a slower decrease in steps spent per episode?
> > >
> > > An agent, like M0, can quickly get to the end yet still fail at turning to the correct direction in the end if they do not learn the rule of accumulating evidence. For example, M0 can quickly navigate through memorization of keep going forward until a certain position, then turn randomly to left or right. This is why we examine both metrics in Fig 2.
> > >
> > > > Since separability of clusters is not a prerequisite for navigation behavior (M3 vs M5 Fig. 9 vs Fig. 2), how can we make sense of the representation to behavior?
> > >
> > > It is true that while the clusters are not separable in the low-dimensional PC space for M3, M3 place cells still exhibit many important biological properties observed in Nieh et al., such as splitter cell phenomenon (i.e., we see choice-specific place cells) and conjunctive place code of position and evidence, as shown in Fig 8, A3 and B. However, the low-dimensional separability implies the downstream processing in cortical regions would be more at ease, as indicated by M5’s superior learning speed in Fig 2A (on bar with M3) and fast navigation shown in Fig 2B in comparison to other models.
> > >
> > > ----
> > >
> > > Please let us know if further clarifications are needed. Thank you again for your time and valuable feedback! We hope that our response and corresponding edits provide sufficient reasons to raise the score.

---

> ### Comment · Reviewer_qnRf · 2024-11-26
> **Complex model where RL does not contribute to representation learning,**
>
> I would like to thank the authors for clarifying some concerns, especially with the analysis. However, I am inclined to maintain my score as I feel the model is unnecessarily complex and the analysis is only performed in one simulation environment.
>
> Unlike the TEM model (Whittington et al. 2020), the Vector Hash model has grid and place codes using bio. plaus. rules which is nice. Furthermore, it has been shown that training just the readout layer of a feedforward/RNN using the temporal difference Hebbian learning algorithm allows agents to learn complex navigation policies  (Kumar et al. 2022 Cerebral Cortex). The authors should have gone with a purely biologically plausible navigation model since the main computation is performed by Vector Hash, and the RL signal is not backpropgated to learn the representations for navigation like in deep RL. This will make the model much more interpretable and easier to train for others to replicate, given the difficulty the authors faced as mentioned in Appendix A.1. The authors should discuss why they chose the RNN + policy network and not simpler alternatives. Will this change the conclusion of the model?
>
> Lastly, the authors should have shown the generality of the model in a variety of 1D and 2D environments instead just one environment.

---

> > ### Author Response · Authors · 2024-11-27
> > **On Biological Plausibility & Generality, and Further Clarifications of Misinterpretation [1/2]**
> >
> > Thank you for your further comments! After carefully reviewing your response, we would like to point out some misinterpretations and seek more clarification respectfully.
> >
> > >  The authors should have gone with a purely biologically plausible navigation model since the main computation is performed by Vector Hash, and the RL signal is not backpropgated to learn the representations for navigation like in deep RL.
> >
> > We thank the reviewer for their thoughtful comments and acknowledge the related work by Kumar et al., 2022, which we have cited in line 133. We agree that a model propagating RL signals fully could offer a good alternative approach.
> >
> > In our work, we combine Vector-HaSH with an RNN policy network trained using the REINFORCE algorithm, a simple and effective RL method. Although the policy network is trained by backpropagation but not the prestructured Vector-HaSH as the reviewer pointed out, RL is still the most common method to model behavioral and neural processes (e.g., Lee et al., 2023; Gershman & Ölveczky, 2020). By incorporating Vector-HaSH, which generates more biologically plausible representations, our framework showcases how these representations can enhance existing RL-based models for studying both behavioral and neural processes.
> >
> > Notably, the accumulating tower task we focus on differs significantly from the Paired Association Spatial Navigation Tasks in Kumar et al., 2022, which involve multiple fixed reward locations and open-space navigation. Instead, our task presents a more complex decision-making problem that requires the agent to infer the relationship between the frequency of tower appearances and the reward location. In this scenario, we believe our use of RL methods is particularly well-suited to capturing the nuanced computations required by the task, such as learning from the interactions with the environment.
> >
> > While we focus here on analyzing neural representations by comparing actual animal experiments with our Vector-HaSH-based models, we acknowledge the importance of further exploring a purely biologically plausible navigation model as future work.
> >
> > > This will make the model much more interpretable and easier to train for others to replicate, given the difficulty the authors faced as mentioned in Appendix A.1.
> >
> > It would be great if the reviewer could clarify the difficulty they referred to. From our understanding, if this refers to the orange text in Appendix A.1, the difficulty we mentioned refers to the pure RNN baseline *without* Vector-HaSH, “For the standalone RNN baselines (M0)...“, which had learning instability. There is no difficulty or additional hyperparameter search conducted for Vector-HaSH + RNN. M1-M5 uses the exact same set of hyperparameters as mentioned in Appendix A.1. Please let us know if we could help with further clarifications.
> >
> > > The authors should discuss why they chose the RNN + policy network and not simpler alternatives.
> >
> > We’d like to politely clarify that the RNN is a policy network by itself and there’s no additional policy network. The RNN is the simplest setup in our view, with just a recurrent layer and a readout layer. Would it be possible to clarify what a simpler alternative could be or if this is a potential miscommunication?
> >
> > > Lastly, the authors should have shown the generality of the model in a variety of 1D and 2D environments instead of just one environment.
> >
> > We appreciate the reviewer’s insightful suggestions. We agree that demonstrating generalizability across multiple environments is always valuable. However, as mentioned in our previous response, we leave this as future work due to the specific scope of our study. Instead, we focused on the learning behavior, neural representations, and the underlying computational roles of multiple regions involved in the accumulating tower task, which serves as a canonical example within the family of spatially embedded decision-making tasks.
> >
> > In neurophysiology experiments, task-related variables—such as recording techniques, animal models, training durations, and specific task parameters—can vary significantly. For instance, Nieh et al. and Pinto et al. both investigated accumulating tasks but focused on different brain regions. Other well-established accumulating evidence tasks, such as the poisson click task (Brunton et al, 2013 and Bondy et al., 2024.), use an auditory sensory input instead of visual sensory inputs and do not have a spatial component. These inherent differences make task generalization more complex. Consequently, most neuroAI and theoretical works demonstrate their methods in a single environment as a proof of concept (e.g., Kumar et al., 2022, Cerebral Cortex; Miller et al., 2023, NeurIPS). We believe our approach is consistent with this established practice in the field.
> >
> > We thank the reviewer again for their thoughtful feedback and will explore extending our approach to other environments in future work.

---

> ### Author Response · Authors · 2024-11-27
> **References [2/2]**
>
> We are keen to hear your thoughts and hope our clarifications have sufficiently addressed the concerns and provided enough reasons for a consideration of raising the score. Please let us know if we could assist with further comments or questions. Thank you again for your time and feedback.
>
> [1] Montague et al., A framework for mesencephalic dopamine systems based on predictive Hebbian learning. J Neurosci., 1996.
>
> [2] Schultz et al., A neural substrate of prediction and reward. Science, 1997.
>
> [3] Lee et al., A feature-specific prediction error model explains dopaminergic heterogeneity. Nature Neuroscience, 2024.
>
> [4] Miller et al., Cognitive Model Discovery via Disentangled RNNs. NeurIPS, 2023.
>
> [5] Pinto et al., Task-dependent changes in the large-scale dynamics and necessity of cortical regions. Neuron, 2020.
>
> [6] Bondy et al., Coordinated cross-brain activity during accumulation of sensory evidence and decision commitment, 2024
>
> [7] Brunton et al., Rats and humans can optimally accumulate evidence for decision-making. Science, 2013.
>
> [8] Gershman & Ölveczky, The neurobiology of deep reinforcement learning. Current Biology, 2020.

---

> ### Comment · Reviewer_qnRf · 2024-11-27
> **Simpler policy network is needed to show EC-HPC is the network integrating temporal information**
>
> A recurrently connected neural network can have readouts as an additional set of nodes and weights, but it is not a given as we can also have an RNN without readouts to learn representations as in Miconi et al. 2017 eLife. I assume there are a total of 3 sets of weights to train in your "policy network": the input to the RNN, the recurrent weights, and the readout weights. Training 3 sets of weights using deep RL allows the model to learn additional representations/features to solve the task, for instance integrating temporal information (Singh et al. 2023 Nature Mach. Int.). Perhaps the EC-HPC network does not capture the temporal dependencies of the Tower Accumulating task, and instead the RNN is the one that is learning to integrate these temporal features to successfully learn a policy. I am starting to think this might be the case.
>
> A clearer way to construct a policy network is to have only one set of weight matrix $W_{ij}$ so that place cell representations or states $s$ directly synapse to the actions $a_j$  following $a_j = \sum_i^N W_{ij} s_i$, similar to having a classification layer in deep networks. This is done by Zannone et al. 2018 Sci. Rep. who uses the Q learning algorithm and Foster et al. 2000 Hippocampus who uses the Actor-Critic algorithm to update $W_{ij}$. Hence, a simpler policy network only has 1 set of weights, that does not need to be trained by back propagation but can be trained by bio. plaus. rules. A model without the RNN but just the policy network will be more convincing to show that the temporal integration of sensory information is done by the EC-HPC network.
>
> I would strongly recommend the authors to evaluate a new model in which the HPC representations feed directly to the policy network, without the RNN, so that $a_{j,t} = \sum_i^N W_{ij} p_{i,t}$. I hope this is clear?

---

> > ### Author Response · Authors · 2024-11-30
> > **Our existing results have addressed reviewer qnRf’s primary concerns**
> >
> > We sincerely appreciate the reviewer’s prompt response and the opportunity to clarify our approach.
> >
> > > I assume there are a total of 3 sets of weights to train in your "policy network": the input to the RNN, the recurrent weights, and the readout weights. Training 3 sets of weights using deep RL allows the model to learn additional representations/features to solve the task, for instance integrating temporal information (Singh et al. 2023 Nature Mach. Int.). Perhaps the EC-HPC network does not capture the temporal dependencies of the Tower Accumulating task, and instead the RNN is the one that is learning to integrate these temporal features to successfully learn a policy. I am starting to think this might be the case.
> >
> > We would like to kindly reiterate that the reviewer’s primary concerns are already addressed by the standalone RNN baseline (Fig. 1, black). The standalone RNN—receiving concatenated sensory, positional, and evidence-velocity inputs—performs poorly, whereas the EC-HPC network (i.e., Vector-HaSH) significantly outperforms it. This result clearly demonstrates that the EC-HPC network, not the RNN, is essential for capturing temporal dependencies and integrating sensory features over time.
> >
> > Furthermore, the reviewer’s concern that the EC-HPC network does not adequately capture temporal dependencies is addressed by our results in Fig. 4 and Fig. 5, B1. Fig. 4 shows that multiple variants of the EC-HPC network exhibit hippocampal firing fields with respect to positions or accumulated evidence, demonstrating that the hippocampus learns the temporal aspects of the task via MEC information flow. Additionally, Fig. 5, B1, illustrates that hippocampal features align with positional information in low-dimensional PC space. Together, these results confirm that temporal integration is intrinsic to the EC-HPC network’s design.
> >
> > Thus, the reviewer’s concerns are already resolved by the current results. If the paper is accepted, we can clarify and further emphasize this aspect in the camera-ready version.
> >
> > > A model without the RNN but just the policy network will be more convincing to show that the temporal integration of sensory information is done by the EC-HPC network.
> >
> > Thank you again for the constructive comments! This is indeed an insightful way to strengthen the learning scheme. If accepted, we think it would be valuable to extend our work to include results using this approach. However, as we [previously elaborated](https://openreview.net/forum?id=9Qfja4ZQW0&noteId=2jgJ95Fr1d),
> > ``` our current approach of integrating RNN with Vector-HaSH is grounded in existing literature, as the most common method to model behavioral and neural processes (e.g., Lee et al., 2023; Gershman & Ölveczky, 2020). By incorporating Vector-HaSH, which generates more biologically plausible representations, our framework showcases how these representations can enhance existing RL-based models for studying both behavioral and neural processes.```
> > Additionally, our use of an RNN abstracts the subcortical and cortical regions (lines 62-63), which are integral to the decision-making process. Modeling these regions as an auxiliary readout, as suggested, would oversimplify their role and deviate from our objective of capturing biologically plausible interactions.
> >
> > As we mentioned earlier, our paper already shows that Vector-HaSH clearly shows the temporal integration of sensory information via Fig 4 and Fig 5, B1. Consequently, although we did not include a fully biologically plausible model but rather combine Vector-HaSH with a RNN policy network learned by the standard reinforcement learning method, the reviewer’s main concern on proving temporal integration by Vector-HaSH is already resolved with current results.
> >
> > Thank you once again for deeply engaging with our work and offering constructive feedback. We hope our further clarifications have sufficiently addressed your concerns and resolved any potential misunderstandings. We also hope that our elaboration on how our current results address the primary issues raised will provide sufficient grounds for reconsideration and raising the score.

---

> ### Comment · Reviewer_qnRf · 2024-12-02
> **Maintain score**
>
> I appreciate the authors' clarifications. The model's strength lies in its rapid map learning, akin to episodic memory. However, I am concerned about its ability to adapt to other neuroscience experiments.
>
> > Additionally, our use of an RNN abstracts the subcortical and cortical regions (lines 62-63), which are integral to the decision-making process. [Author Response]
>
> The authors mentioned that M0: RNN + RL network is not able to solve the task (Fig. 2), and the grid+place cell network added to the RNN improves learning. Hence, it seems contradictory to me that the authors emphasize on the importance of using an RNN to model decision-making again. As mentioned before, it will be much clearer to show that a grid-place-policy network, without the RNN, improves policy learning. Please see the list of references given before that shows a simpler model and grounding to neuroanatomy, which will make it easier to adapt the current model to other simulations. To me, overcomplicating a model to show multi-region interaction is obscuring the computational understanding offered by simpler alternatives.
>
> > Such an understanding could reveal how cognitive maps could be leveraged not only to navigate physical spaces but also to guide cognitive decisions. [Pg. 1, Line 51]
>
> > We believe the current scope of our work is well contained. [Author Response]
>
> I would like to re-emphasize that the current scope of the experiments to make the claim that the 'cognitive' map learned by the model improves policy learning is insufficient. The authors should have shown at least 1 additional simulation in a different environment perhaps with obstacles, or a non-spatial but cognitive task to further validate that the representations learned by the model is general.
>
> Another suggestion is to show that the model replicates the rapid learning seen in Tse et al. 2007 Science by learning schemas. Some modeling papers include Hwu & Krichmar 2020 Biological cybernetics and Kumar et al. 2021 arXiv 2106.03580.
>
> The current model is capable of improving our understanding on a range of interesting cognitive questions e.g. schemas, beyond the tower accumulation task. I urge the authors to consider these suggestions to strengthen the model's robustness.

---

### Official Review · Reviewer_LP6J · 2024-11-03

**Soundness:** 2
**Presentation:** 3
**Contribution:** 3
**Rating:** 5
**Confidence:** 3

**Summary:**

This paper models the entorhinal-hippocampal-neocortical circuit in the brain using a Vector-Hash + RNN model and trains it with reinforcement learning to accumulate sensory evidence and make decisions. The authors test several model hypotheses, including whether the MEC receives sensory evidence, whether sensory information and spatial information are co-represented, and whether the HPC binds sensory and spatial information through associative memory. Their results indicate: (1) in terms of task performance, inputting both spatial and sensory information to the MEC improves learning speed; (2) in terms of neural experiment interpretation, the mixing of sensory and spatial information in the MEC results in cotuning of evidence and position in HPC place cells, as well as choice-specific firing patterns, similar to experimental findings. These results suggest that integrating multimodal information in the MEC for abstract path integration may be a biological strategy for decision-making tasks.

**Strengths:**

The first strength lies in the novelty of the model architecture. This paper introduces a new framework for modeling the entorhinal-hippocampal-neocortical loop, combining an RNN trained with gradient backpropagation with a hand-designed, Hebbian-learning-based Vector-Hash module. This approach offers a unique testbed for exploring hypotheses about brain region function and information transfer, providing valuable insights for future research in this area.

Another strength is the linkage between experimental neural findings and the model’s performance on specific tasks. The paper connects cotuning of evidence and spatial location, as well as choice-specific place cell firing, with efficient spatial navigation and rapid learning in the network, potentially advancing brain-inspired intelligence research.

**Weaknesses:**

The primary limitation is the oversimplification of the biological network and the lack of rigorous model comparisons, which affects the credibility of the findings. First, the model reduces the entorhinal-hippocampal circuit to a Vector-Hash model, which, while capable of explaining associative and episodic memory functions attributed to these areas, significantly simplifies the actual biological network. For instance, CA3 in the hippocampus has extensive recurrent connections, whereas the Vector-Hash model lacks internal connections, reducing the hippocampus to an intermediary layer connecting MEC and LEC. This simplification may have critical effects when studying how the model produces co-tuned place cells and choice-specific firing since it removes internal computation within the hippocampus, relying instead on upstream grid cell networks.

Secondly, when proposing joint integration of evidence and spatial velocity in the MEC as a means to enhance learning, the paper lacks a fair comparison of different models' capabilities. For a balanced evaluation, it would be essential to compare the performance of a standalone RNN (without Vector-Hash) receiving both physical velocity and sensory evidence, with an equivalent number of neurons. This is a minimal comparison proposal to clarify Vector-Hash’s role, although it alone may not be fully sufficient for rigorous validation.

**Questions:**

Overall, I appreciate the paper’s attempt to model the entorhinal-hippocampal-neocortical circuit, especially combining classic theoretical models with RNNs. However, I am somewhat skeptical about the reliability of the conclusions, which impacts my overall assessment. I have several questions, including both major and minor ones, related to the results and model details.

**Major Questions:**
1. In hypothesis M2, where sensory evidence is bound to the HPC, and downstream RNN receives HPC activity as input, why does the network fail to learn effectively, performing worse than an RNN receiving sensory evidence directly? Since spatial location is also useful in this task, as the agent needs spatial navigation for reward acquisition, the paper suggests that coupling both inputs makes it difficult for the downstream RNN to decouple them. Could the authors elaborate on this point?

2. According to Figure 2B, the RNN in M0 can perform efficient navigation, even outperforming the network in M3. This seems counterintuitive, as M3 includes additional spatial location information, which is crucial in a spatial navigation task. Could the authors discuss this in more detail?

3. In the disjoint integration model (M4), no co-tuned place cells emerge in the HPC. Under this model framework, is it because separate grid cell modules lead to orthogonal representations of evidence and position? If so, a natural prediction would be that "evidence cells," tuned purely to evidence spacing, might appear in the HPC. Is this observed in the actual results?

4. I believe that omitting recurrent connections within the HPC contributes to the lack of co-tuned cells when evidence and spatial location are combined in the HPC. If the HPC were replaced with a standard RNN receiving both MEC spatial input and LEC evidence input, would co-tuning place cells still emerge even if the MEC does not receive evidence or integrates evidence and position disjointly? I recommend the authors investigate this, as it may determine whether the findings are artifacts due to biological oversimplification.

**Minor Questions:**
1. In Equation (1), the operation CAN(.) is not clearly defined in the main text or the Appendix. As CAN is a classic, well-studied dynamical model with various implementations, could the authors clarify the specific form used here?

2. In the joint-integration model, how exactly are evidence and position information combined before being input to the MEC? Is it a simple summation?

---

> ### Author Response · Authors · 2024-11-18
>
> *Thread [1/3]*
>
> Thank you very much for your detailed and thoughtful feedback! We highly appreciate your time helping us to improve the quality of our work. Here we address your comments in order, in addition to modifications made to the manuscript, including Fig 1C, 1E, Discussion, and other minor edits in blue for added clarity on methodology and result interpretation.
>
> **Regarding weakness:**
>
> > For instance, CA3 in the hippocampus has extensive recurrent connections…This simplification may have critical effects when studying how the model produces co-tuned place cells and choice-specific firing since it removes internal computation within the hippocampus, relying instead on upstream grid cell networks.
>
> The consideration of CA3 recurrence is a very valid point, but we didn’t include it due to the scope of our work. We have added an additional paragraph discussing this limitation to our Discussion section. For TLDR; reasons are several-fold, such as the reasons being brought up by reviewer bpUy (Question 1) that we focus on entorhinal-hippocampal interactions due to its importance to cognitive map theory and spatial navigation. Furthermore, considering CA3 recurrence is, in fact, complementary to our work, as testing the counterfactuals of CA3 recurrence would still involve M0-M5, and their variants. This extra consideration can be easily tested using our framework as detailed in the additional paragraph in Discussion, but it adds an extra layer of complexity, making it outside the scope of this work. Since our work generates a straightforward, falsifiable prediction, we are currently collecting neurophysiological data to verify our predictions–the result will directly inform whether other mechanisms play a role, such as CA3 recurrence, and can be easily investigated in our current framework as described in lines 518-522.
>
> > For a balanced evaluation, it would be essential to compare the performance of a standalone RNN (without Vector-Hash) receiving both physical velocity and sensory evidence, with an equivalent number of neurons.
>
> Thank you for pointing this out. M0 is a standalone RNN, intaking sensory only. All model variants have RNNs of the same size (hidden size of 32, indicated in Appendix A.1). The point of including M0 is to provide a baseline when an abstractor of velocity information, i.e., VectorHaSH, is absent. If the standalone RNN instead receives abstracted velocity information, in contrast to other models that receive hippocampal representations, the task would be trivial. We could be misunderstanding the proposal, e.g., did you mean to have an RNN of size 32 to receive sensory vectors concatenated with MLP-abstracted evidence velocity, instead of just the sensory vectors in the current setup? Currently, in M0-M5, no RNN receives velocity information directly. Any clarification would be appreciated, thank you!

---

> > ### Author Response · Authors · 2024-11-18
> >
> > *Thread [2/3]*
> >
> > **Regarding major questions:**
> >
> > > In hypothesis M2…why does the network fail to learn effectively, performing worse than an RNN receiving sensory evidence directly? Since spatial location is also useful in this task…
> >
> > Spatial information is indeed useful to the task, but not as critical as the ability to accumulate evidence correctly. For example, an agent could navigate quickly simply due to memorization, and fails the task ultimately if it makes a turn to left or right randomly in the end. The difficulty for downstream decoupling can be a consequence of M2 missing evidence firing fields (Fig 8, A2), potentially due to the difficulty of disentangling evidence from sensory information through simple projection, without a structured abstractor i.e., grid cells. This is in contrast to well-performed M3-M5 (Fig 8, A3, and Fig 4, B2, B3).
> >
> > > According to Figure 2B, the RNN in M0 can perform efficient navigation, even outperforming the network in M3. This seems counterintuitive, as M3 includes additional spatial location information, which is crucial in a spatial navigation task. Could the authors discuss…
> >
> > Intuitively, grid code provides rigid information of an environment and the task at hand, such as the spatial information you mentioned. On the other hand, non-grid sensory input to HPC (or sensory to RNN) is potentially helpful to capture the nuances in the environment, such as where the wall is (marked as ‘-1’, hinting the decision region). This source of nuances is present in M0, M4, and M5, and we indeed observe fast navigation in both M0 and M5. M4 does not exhibit fast navigation, which can be due to other confounds such as its disjoint grid code. For added clarity, we added lines 358-360 in blue. Thank you for your feedback!
> >
> > > In the disjoint integration model (M4), no co-tuned place cells emerge in the HPC. Under this model framework, is it because separate grid cell modules lead to orthogonal representations of evidence and position? If so, a natural prediction would be that "evidence cells," tuned purely to evidence spacing, might appear in the HPC. Is this observed in the actual results?
> >
> > Thank you for pointing this out. We did not observe HPC cells tune to evidence or position only in M4. This can be inferred from Fig 6D directly, for example, a cell only tuned to evidence should have a mutual information value on the X=Y line in D1, since the positional information (Y) should not matter. To further make sure this is not an artifact due to EC projection to HPC, we ran M4 without an activated EC-HPC pathway. We observed a similar mutual information pattern as Fig 6D, and similar ExY fields shown in Fig 3 top row, where the firing fields are rigid and multiple ExY peaks can exist in one cell, but we do not see any continued stripe with respect to one axis like those in Fig 7A. We did not identify choice-specific cells in either case (a visualization for M4 is in Fig4, A2).
> >
> > > I believe that omitting recurrent connections within the HPC contributes to the lack of co-tuned cells when evidence and spatial location are combined in the HPC…I recommend the authors investigate this, as it may determine whether the findings are artifacts due to biological oversimplification.
> >
> > Thank you for the proposal, and this is the potential next step of our work, depending on the experimental verification result. We have discussed our reasons for exclusion when addressing paper weakness above (e.g., our predictions are experimentally testable and falsifiable) and in the new paragraph added to Discussion (lines 516-526). The current findings would not be affected and are not the outcome of simplification, as it carefully tests the effect of grid code on place cell representation and behavior performance. Testing the effect of CA3 recurrence would still require various ablation studies including our M0-M5, and their variants. The role of CA3 recurrence can be complementary, and is definitely not mutually exclusive to the current prediction.

---

> > > ### Author Response · Authors · 2024-11-18
> > >
> > > *Thread [3/3]*
> > >
> > > **Regarding minor questions**
> > >
> > > >  As CAN is a classic, well-studied dynamical model with various implementations, could the authors clarify the specific form used here?
> > >
> > > Thank you for the clarifying question. The velocity inputs to each grid module updates the grid phases through path integration, following Vector-HaSH in Chandra et al., 2023, akin to Burak & Fiete, 2009. We have also added Fig 1C for added clarity on the disjoint/joint grid module coding scheme, and elaborated in lines 198-200 and Appendix A.1.
> > >
> > > > ​​In the joint-integration model, how exactly are evidence and position information combined before being input to the MEC? Is it a simple summation?
> > >
> > > Different task variables each utilize a different axis of a grid module, also addressed in the new Fig 1C and added lines 198-200, along with Appendix A.1.
> > >
> > > ----
> > >
> > > Please let us know if further clarifications are needed. Thank you again for your time and valuable feedback! We hope that our response and corresponding edits provide sufficient reasons to raise the score.

---

> ### Comment · Reviewer_LP6J · 2024-11-24
>
> Thank you for your response and for taking the time to update the manuscript. However, after reviewing your reply, I feel that my primary concerns regarding the weaknesses of the study have not been fully addressed. I would like to elaborate further, especially considering the importance of these points to the validity of your core conclusions.
>
> 1.	On the exclusion of CA3 recurrence:
>
> One of the key contributions of your paper is the prediction that joint integration of velocity and evidence in grid cells is critical for producing co-tuning of place cells in the hippocampus. However, as I mentioned in my review, this conclusion may be significantly influenced by the structural simplifications in your model, particularly the omission of CA3 recurrent connections. The hippocampus, as a central component of the cognitive map, is widely understood to integrate information from multiple sources, and the recurrent connections in CA3 are thought to play an essential role in this integration.
> While I understand that the scope of your work focuses on entorhinal-hippocampal interactions, simply stating that your prediction is falsifiable does not justify omitting CA3 recurrence from your analysis. Instead, it is essential to explain why the exclusion of CA3 recurrence does not compromise your prediction or, at the very least, discuss how it might influence the interpretation of your results. This is especially crucial given that your model already incorporates projections from the lateral entorhinal cortex (LEC) to the hippocampus, which should facilitate evidence integration in the hippocampus.
>
> Considering the central role of this prediction in your study, a more thorough justification or analysis is necessary to demonstrate that the lack of CA3 recurrence does not undermine your conclusions. Otherwise, the validity of this prediction remains questionable and risks being an artifact of the model's structural oversimplification.
>
> 2.	On the role of a standalone RNN:
>
> In your manuscript, the first section of the Results is titled "Joint Integration of Position and Evidence in MEC Induces Rapid Learning". This title implies that introducing joint coding of velocity and evidence in the Vector-Hash model (e.g., M3 and M5) leads to faster learning compared to a standalone RNN (e.g., M0). Indeed, Figure 2A shows that M3 and M5 achieve higher accuracy more quickly than M0. However, I believe this comparison is not entirely fair for the following reasons:
>
> •	The RNN in M0 has significantly fewer total neurons than the combined RNN and Vector-Hash modules in M3 and M5, which inherently puts M0 at a disadvantage in learning capacity.
>
> •	The RNN in M0 receives only sensory information as input, whereas the Vector-Hash models also encode velocity information, which is highly task-relevant.
>
> To make a fair comparison, I suggest including an additional condition where an RNN with the same number of neurons as M3 or M5 receives both velocity and sensory inputs. This would allow readers to directly compare the learning capabilities of the standalone RNN and the proposed RNN + Vector-Hash model.
>
> If your intention is not to claim that the RNN + Vector-Hash model learns faster than a standalone RNN, then the phrase "induces rapid learning" should be clarified in the manuscript. Specifically, the comparison target for this claim should be explicitly stated to avoid potential misinterpretations.

---

> ### Author Response · Authors · 2024-11-25
>
> Thank you very much for the insightful and timely elaborations! We now better understand the concerns, and have conducted relevant experiments.
>
> In addressing concern (1) of CA3 recurrence, we have added Fig 11 and Fig 12 to Appendix F. These results serve as a proof of concept, that adding CA3 recurrence to M2 (position only grid code + mix p) or M4 (disjoint grid code + mix p) is not sufficient to drive experimentally observed phenomena in place cells, showing such a model simplification does not undermine our conclusions drawn in the main text. We have modified the relevant text in Discussion accordingly, highlighted in orange in lines 517-525, elaborating the implication and future directions.
>
> In addressing concern (2) of baseline comparison, we have included an additional standalone RNN baseline to Fig 2 in black. We additionally scaled up the original standalone RNN baseline in blue. Specifically, the standalone RNNs now have a hidden size of 32 + Ng + Np + Ns (same number of neurons as RNN + Vector-HaSH in M5). The RNN takes in sensory info, while for the black variant, it is concatenated with position velocity and evidence velocity (predicted by MLP). The previous conclusion remains the same, except we observe learning/training instability in these large-size RNNs--the implementation details are elaborated in Appendix A.1.
>
> Thank you again for your time and valuable feedback! Please let us know if further clarifications are needed in addressing your concerns. We hope that our response and corresponding edits provide sufficient reasons to raise the score.

---

> > ### Comment · Reviewer_LP6J · 2024-11-30
> >
> > Thank you for your thorough follow-up response, as well as the additional experiments and clarifications. I appreciate the effort you have put into addressing my concerns.
> >
> > Regarding the first concern about CA3 recurrence, I acknowledge the inclusion of new experiments (Fig 11 and Fig 12 in Appendix F) and the accompanying discussions in the manuscript. However, I remain unconvinced that the role of CA3 recurrence in generating conjunctive place cells can be fully excluded based on your current results. My primary concern is that the learning rule employed in your model is limited to Hebbian learning, which is inherently simplistic and not sufficient for solving many tasks that require more complex associative computations. This limitation might restrict the potential of CA3 recurrence to manifest its functional role in your simulations. I encourage further exploration in future work with more sophisticated learning rules to fully address this question.
> >
> > On the second concern about baseline comparisons, I am satisfied with the additional standalone RNN baselines you included in Fig 2 and the related implementation details. The new results provide a more comprehensive view of the model's capabilities, and I appreciate the inclusion of these comparisons.
> >
> > Given the current stage of revisions, I do not intend to propose any additional changes. I am considering raising my score from 5 to 6 but have not yet made a final decision. I will submit my evaluation before the deadline.
> >
> > Thank you again for your time and dedication to addressing my feedback.

---

### Official Review · Reviewer_bpUy · 2024-11-03

**Soundness:** 4
**Presentation:** 3
**Contribution:** 3
**Rating:** 8
**Confidence:** 4

**Summary:**

This paper proposed a multi-region brain model for the hippocampo-entorhinal-neocortical circuit in a spatially embedded decison-making task. The purpose of the model is to understand the neural mechanism underlying the conjunctive encoding of position and evidence in the hippocampus in the accumulating tower task. By simulating the task as a RL problem, this paper demonstrates that 1) conjunctive encoding of position and evidence in the MEC and 2) non-grid sensory input from EC to HPC are necessary for the conjunctive representation in the HPC. The conclusion is reached by performing a rigorous testing of mutiple alternative hypothesese and comparing the model representations with hippocampal representations obtained experimentally. Overall, this is a solid paper that asks and answers a very interesting neuroscience question with well-designed experiments making use of an existing model.

**Strengths:**

- Rigorous testing of alternative hypotheses to the proposed one: I really appreciate the examinations of other possible mechanisms of conjunctive hippocampal encoding (Table 1 and 2) in the paper, which is sometimes missing in many computational neuroscience papers. These examinations not only serve as an ablation study that makes the results more robust but are also biologically interpretable, which definitely strengthens the argument of the paper
- Reference to experimental results (i.e., Nieh et al., 2021): Many computational neuroscience works assume the reader, usually from a computational background, understands the experimental setup they simulate very well. In such cases the reader has to carefully go through the original experimental work to fully understand the context of the model. This paper does quite a good job in this regard by clearly presenting the simulated task very early and referring to the experimental results from Nieh et al. 2021 frequently.

**Weaknesses:**

- I didn't find any significant weakness of this paper that can be summarized under general topics. There are some detailed questions/unclarity I wish the authors could address, which I noted under Questions.

**Questions:**

- Why do the authors study the two specific reasons i.e., conjunctive encoding of space and evidence in the MEC and sensory inputs, other than other possible mechanisms underlying the conjunctive hippocampal encoding? Are there any experimental evidence demonstrating that these two are the most likely/important ones? I understand these two reasons are related to the interactions between (M)EC and HPC, but is it possible that the conjunctive representation in HPC is also due to internal reasons e.g., interactions between HPC subregions? The authors might want to add a sentence or two to highlight why they chose to study these two reasons specifically.
- The grid-cells module in the model takes pos and evidence as inputs. I understand that the evidence is a MLP-projected representation of the sensory inputs, which the authos presented quite clearly in lines 215-222. However, it is unclear how position is represented and input into the grid cells. Is it encapsulated in the CAN module?
- The MLP module models how evidence is extracted from sensory inputs, and from Fig1B, it seems to receive sensory inputs directly from non-grid EC. Biologically, is this a known mechanism? Could it be receiving inputs directly from sensory regions? Which brain region and what neural mechanism does this MLP correspond to? Maybe something to include in the Discussion.
- How did you achieve disjoint grid cell encoding of space and evidence in your model? i.e., computationally and mathematically, how did you make a difference between joint and disjoint grid cell codes? You may want to include this in section 3.2 Model Setup, or in the appendix.
- In line 309 'we show that our multi-region brain model (M4)' - you mean M5 right?
- Fig3 left: if space allows, you may want to include Fig1d from Nieh et al. 2021 to make a contrast to right-choice-selective place cells, as readers unfamiliar with their work might struggle to understand what this diagram means.
- Section 5.3.1: I feel this whole subsection is a bit unclear. I wish you can check/clarify the following points:
    - I think this section aims to demonstrate only M5 shows separable clusters of tasks variables, but the title says 'only joint integration model exhibits...' - both M3 and M5 are joint integration model, but M3 doesn't have activated EC pathway right? So I guess what you want to say is 'only joint integration model with activated EC pathway exhibit...'? This is also related to the fact that you are showing in Fig5 that both M4 and M5 are action-separable, but M4 is a disjoint model.
    - On line 431, you refer to Fig 5 A1/A2 for separable clsuters. I guess you should refer to B1/B2?
    - On line 467, you mentioned that you did not observe separability in accumulated evidence. Can you show some example, at least in the appendix?
    - I also feel the whole Fig5 could improve with a re-wording of the caption clarifying which model (M3,4,5) each row (A, B) and each column (1,2,3,4) correspond to. Currently it is not very clear.

---

> ### Author Response · Authors · 2024-11-18
>
> *Thread [1/1]*
>
> Thank you very much for your time, and insightful, detailed comments. We truly appreciate your kind words, and your valuable time helping us to improve the quality of our work.  Here we address your questions and feedback in order, with modifications applied to Discussion, Fig 1, Fig 5 caption; we added Fig 10, and additional text to Method (in blue):
>
> > …is it possible that the conjunctive representation in HPC is also due to internal reasons e.g., interactions between HPC subregions? The authors might want to add a sentence or two to highlight why they chose to study these two reasons specifically.
>
> Thank you–it is a very valid point regarding the internal interactions within HPC subregions. Aside from the reasons you mentioned, we did not include it for reasons we just added to the Discussion section of the manuscript (in blue, lines 516-525). TLDR; being the consideration of e.g., CA3 recurrence would be complementary to our current work. And, given we have a falsifiable and straightforward prediction of conjunctive grid code, we are collecting neurophysiological data to verify this prediction directly. The result will directly inform whether other mechanisms play a role, and can be easily investigated with our current framework with methods described in lines 518-522.
>
> > However, it is unclear how position is represented and input into the grid cells. Is it encapsulated in the CAN module?
>
> Positional velocity is represented as 0 (stuck) or +1 (forward) without the use of MLP given backward-moving is not task-relevant (Nieh et al., 2021, see behavioral training), though it is possible to use an MLP as well. It will not affect the results. Generally, the velocity inputs to each grid module updates the grid phases through path integration, following Vector-HaSH implementation in Chandra et al., 2023, akin to Burak & Fiete, 2009. For added clarity, we added a grid code schematic of how velocity is represented with respect to inputs in Fig 1C and elaborated in lines 198-200 and Appendix A.1.
>
> > The MLP module models how evidence is extracted from sensory inputs, and from Fig1B, it seems to receive sensory inputs directly from non-grid EC…what neural mechanism does this MLP correspond to? Maybe something to include in the Discussion.
>
> In our model, the sensory input and the encoding of non-grid EC are technically the same as a consequence of our setup (lines 215-217). The MLP predicts velocity from sensory directly instead of from non-grid EC (lines 200-201). We agree our schematic was misleading, and thank you a lot for pointing this out! We have modified Fig 1B and 1E correspondingly. We have also added to the Discussion our justification of using an MLP (lines 526-532 in blue), in addition to the earlier mention of its potential biological implication in lines 227-228.
>
> > How did you achieve disjoint grid cell encoding of space and evidence in your model? i.e., computationally and mathematically, how did you make a difference between joint and disjoint grid cell codes? You may want to include this in section 3.2 Model Setup, or in the appendix.
>
> Thanks for pointing this out! We added Fig 1C for added clarity and elaborated in lines 198-200 and Appendix A.1 (as mentioned above in pt. 2). The joint/disjoint grid coding scheme is a manual setup so we can have a careful, controlled comparison of both possibilities. Please let us know if it’s still unclear, and we are happy to clarify more.
>
> > In line 309 'we show that our multi-region brain model (M4)' - you mean M5 right?
>
> Yes, thanks. This is now fixed.
>
> > Fig3 left: if space allows, you may want to include Fig1d from Nieh et al. 2021 to make a contrast to right-choice-selective place cells, as readers unfamiliar with their work might struggle to understand what this diagram means.
>
> Yes, it was a part of the consideration, but we removed it since Nieh et al. confirm that Fig1e was the experimental outcome, so we didn’t want to confuse readers with the other possibility that was ruled out experimentally.
>
> > Section 5.3.1: I feel this whole subsection is a bit unclear. I wish you can check/clarify the following points…
>
> Thank you for pointing this out and for your constructive feedback! We have made modifications with respect to each point. Specifically, we modified the section 5.3.1 title and fixed the relevant typo. We have included the separability of accumulated evidence in Figure 10. We have also reworded Fig 5 caption.
>
> ----
>
> Please let us know if further clarifications are needed. Thank you again for your time and valuable feedback!

---

> > ### Comment · Reviewer_bpUy · 2024-11-24
> >
> > Thank you for your clarification. This paper is much more complete and clearer now. I don't have any more concerns. Good luck :)

---

### Official Review · Reviewer_TiD8 · 2024-11-04

**Soundness:** 2
**Presentation:** 2
**Contribution:** 2
**Rating:** 3
**Confidence:** 3

**Summary:**

The authors extend the Vector-HaSH model (Chandra et al. 2023), a recently developed neural network model inspired by the hippocampal (HPC)-entorhinal cortex (EC) circuit, to include an MLP-based model of the sensory neocortex and an RNN module to model the brain circuits involved action selection and reinforcement learning. The authors demonstrate the model’s ability to rapidly learn the accumulating tower task, and to generate place cell maps that resemble those observed in the brain (based on data from Nieh et al. 2021). Drawing comparison across 5 models with different architecture and grid and place cell coding schemes, the authors conclude that location-evidence conjunctive coding in grid cells and non-grid EC inputs to the HPC are important for rapid learning and the formation of conjunctive maps in the HPC.

**Strengths:**

Understanding how the HPC-EC circuit interact with other brain regions in spatial cognition tasks is an important open question. Extensive modeling effort has been focused on the HPC-EC circuit, while sophisticated models linking HPC-EC with other brain regions remain lacking. Hence this study is an important step towards the right direction.

**Weaknesses:**

1)	The neocortex is modeled as a MLP and appears to the only channel by which sensory input enters the EC-HPC circuit. Since the MLP module plays the important role of extracting the evidence velocity and conveying to the Vector-HaSH module, it is unclear comparison with Models 0-2, which do not have the MLP module, is fair.
2)	It is not entirely clear how the different coding schemes of the grid cells (position only, disjoint pos+evi, joint pos+evi) are incorporated into the model. In line 309, the authors claim that the joint coding scheme emerged in M4, while Table 2 seems to claim otherwise? Do different grid codes in Models 3-5 emerge due to difference in the architecture? Or other network parameters?  It would be greatly helpful if the authors can provide more implementation details clarifying how the models differ from each other, which would also help distinguish which observed phenomena arise by design and which emerge due to optimizing for task performance?
3)	The authors show that M5, with a joint grid code and non-grid input to HPC, yielded faster learning. However, there lacks an intuitive insight on why this is the case. Could the author provide a mechanistic insight and some supporting analyses?
4)	The comparison between model dynamics and biological data appear largely qualitative. For example, the PCA results show some visual clusters in B but less so in A, but how can we be sure there are no separable clusters in high dimensional space?
5)  Other implementation details, such as how the RNN is trained in an RL framework, is lacking.

**Questions:**

1) Could the authors clearify how the 5 models differ in architecture, external input, and/or hyperparameter?
2) Could the authors provide mechanistic intuition on why the joint grid emerged (or is incorporated by design) in M3 &5? (please also clairfy on the claim regarding M4 in line 309-310)

---

> ### Author Response · Authors · 2024-11-18
>
> *Thread [1/2]*
>
> Thank you very much for your valuable time and feedback in helping us improve our work! Here we address the weaknesses and questions. Additionally, to enhance the clarity of our methodology, we have added additional figures, Fig 1C, 1E, Fig 10, and further elaboration in lines 198-200 and Appendix A.1.
>
> > The neocortex is modeled as a MLP and appears to the only channel by which sensory input enters the EC-HPC circuit.
>
> We’d like to kindly clarify—sensory information in non-grid EC layer (Fig 1B green) can also enter the circuit (M2, M4, M5; detailed in Table 1, 2).
>
> > Since the MLP module plays the important role of extracting the evidence velocity and conveying to the Vector-HaSH module, it is unclear comparison with Models 0-2, which do not have the MLP module, is fair.
>
> We believe the comparison is rigorous and logical. The main goal of the paper is testing hypotheses of neural computation, as shown in Table 1–the point of including M1, M2 is to rigorously span the entire hypothesis space of how conjunctive coding in place cells arise. To put Table 2 more clearly, the inclusion of M1-2 follows a simple logic, where the hypothesis space is
>
> | Evidence from EC | Evidence from MEC | Corresponding model of the hypothesis |
> | ---------------- | ---------------- | ------------------------------------- |
> | False            | False            | M1                                    |
> | True             | False            | M2                                    |
> | False            | True             | M3                                    |
> | True             | True             | M4, M5                                |
>
> The point of including M0 is to provide a baseline when an external integrator of velocity information, i.e., VectorHaSH, is absent.
>
> > It is not entirely clear how the different coding…joint coding scheme emerged in M4…Do different grid codes in Models 3-5 emerge due to difference in the architecture? Or other network parameters?
>
> Thank you for pointing this out! Line 309 (now line 304) was a typo, and is now fixed. The grid coding scheme is carefully controlled by us to test how different coding schemes affect the downstream computation, e.g., place cell representation, task performance.
>
> To add clarity, we added Fig 1C and 1E in the latest version to illustrate the differences between the coding scheme and model architecture, and elaborated in lines 198-200 and Appendix A.1. As Fig 1C shows, the disjoint grid code means each grid module only gets velocity input of one variable, represented by one axis of the 2D representation space; the joint grid code means both axes of the 2D space are utilized in each grid module. Both coding schemes are capable of providing distinct code for different states of position and evidence, but the downstream representation and performance turns out to be different.
>
> The above also addressed both questions you had. The hyperparameters such as network size and learning rate are shared across models–these details are stated in Appendix A.1. Please let us know if anything else is still unclear, and we are more than happy to elaborate.

---

> ### Author Response · Authors · 2024-11-18
>
> *Thread [2/2]*
>
> > Could the author provide a mechanistic insight and some supporting analyses?
>
> Intuitively, grid code provides rigid information of an environment and task at hand, but non-grid input to HPC is potentially helpful to capture the nuances in the environment, such as where the wall is (hinting the decision region). For added clarity, we added lines 357-359 in blue. Thank you for your feedback!
>
> The mechanistic insight of M5’s faster learning can be supported by the analysis in Fig 4, A3, showing splitter cell phenomenon, in which place cells show differential activity based not only on spatial location but also on additional context (”choice-specific”). This is also supported by analyses of Fig 5, B, and quantitative measure in the new Fig 10, showing the hippocampal representation is (visually) separable in low dimension under joint grid code and activated EC pathway.
>
> > The comparison between model dynamics and biological data appear largely qualitative…how can we be sure there are no separable clusters in high dimensional space?
>
> We focus on low-dimensional representations, especially given the task is low-dimensional and Nieh et al. found the experimental hippocampal population activity is constrained to a low-dimensional manifold. High dimensional separability is very likely, but even so, given M5 is separable in low dimension while other variants are not, this characteristic alone underscores the computational advantage of mechanisms implied by M5. We have modified the wording in section title of 5.3.1 to reflect our intention. We also appreciate your feedback on more quantitative measures, and have added scree plots of M1-M5, i.e., cumulative variance explained by numbers of PCs, to the Appendix Fig 10 (row 1). Quantitatively, M5's first 2 PCs of hippocampal activity can explain 68% of variance, while M3 only explains <20%, similar to other models. Though the comparison of place cell coding is qualitative-oriented in Fig 4, the place cells are selected through a quantitative mutual information analysis (Appendix B).
>
> > Other implementation details, such as how the RNN is trained in an RL framework, is lacking.
>
> The implementation details are mentioned: the RL training was briefly described in line 248, “which is an action-selection RNN policy trained through policy gradient under reinforcement learning…” which points to a more detailed description in the Appendix A (line 249), “Please refer to Appendix A.3 for what one step by the agent in the environment entails among the involved brain regions.” These are also mentioned in line 312.
>
> ----
>
> Please let us know if further clarifications are needed. Thank you again for your time and valuable feedback! We hope that our response and corresponding edits provide sufficient reasons to raise the score.

---

### Official Review · Reviewer_L6Mf · 2024-11-04

**Soundness:** 3
**Presentation:** 2
**Contribution:** 1
**Rating:** 3
**Confidence:** 4

**Summary:**

The authors introduce a modular architecture which integrates position and sensory evidence to create a RL agent capable of performing a common experimental neuroscience task. They specifically show that integration of sensory information in the module responsible for grid cell activity is essential for performance of this task.

**Strengths:**

The methodology described in section 3.2 seems to accurately match the previously published methodology. The methods relating to learned representations (5.2-5.3.1) show a strong similarity to experimental properties such as place-field location biases. The conjunctive tuning curves (eg: Figure 3 & 7) are particularly convincing.

**Weaknesses:**

-	Overall, it is difficult to follow the logic of the paper. The introduction begins with a very specific example of neurophysiological findings and then seeks to justify the choice of Vector-HaSH. However, the justification for this starting point is extremely weak (line 122-123). Given that the proposed model is a slight modification of this previously published work, by the introduction of an MLP, there must be a much stronger argument for the validity of the chosen base model.
-	Relatedly, the model is a small modification of the previously published vector-hash, followed by detailed investigation of performance and similarity of learned representations to experimental findings. The work therefore seems more appropriate for a neuroscience venue. In the current format, it is unclear what the implications for representation learning are.
-	Section 2.3 is an odd aside, and the text does not seem to link it to the proposed model or findings at hand.

**Questions:**

- How can these findings be more directly linked to representation learning and machine learning at large? Beyond explaining specific phenomenon in neuroscience, there do not appear to be any general ML findings. While there are paragraphs (2.3 and discussion) claiming that neuroscience findings can guide neuro-inspired AI, no concrete predictions or general findings are made.

**Details Of Ethics Concerns:**

Figure 1 A-B is directly copied from Chandra 2023. This may be either a copyright concern or unintentionally reveal the authors of the paper.

---

> ### Author Response · Authors · 2024-11-18
>
> *Thread [1/2]*
>
> Thank you very much for your valuable time and feedback in helping us improve our work! Here we address the weaknesses and questions respectively, in addition to the revision made in Introduction to emphasize on the ML relevance of our work (lines 65-70, in blue)
>
> **Regarding weakness**
>
> > Overall, it is difficult to follow the logic of the paper. The introduction begins with a very specific example of neurophysiological findings and then seeks to justify the choice of Vector-HaSH.
>
> We would like to respectfully point out that the neurophysiological findings we reference (Nieh et al., 2021, Nature) are not just specific examples but are representative of broader and well-established principles in cognitive map theory, which is foundational for our study (Section 2.1). We chose these findings because they offer robust experimental evidence regarding place cell conjunctive tuning of physical and cognitive variables, which serves as a strong ground truth for testing hypotheses in our model. By aligning our simulations with these generalizable experimental results (Fig 3, 4), we demonstrate the biological relevance and predictive power of our ML-oriented approach. Therefore, the introduction of these findings is not only logical but essential for grounding the context of our work.
>
> > the proposed model is a slight modification of this previously published work, by the introduction of an MLP
>
> > Relatedly, the model is a small modification of the previously published vector-hash...
>
> We would like to kindly point out that our method is not just a simple modification of Chandra et al., (2023). While it is true that our approach to modeling the entorhinal-hippocampal interactions is based on Vector-HaSH, it's not the entirety of our multi-region model (Section 3.1, Fig 1B, D, E; e.g., we introduce a new framework for modeling the entorhinal-hippocampal-neocortical loop, as highlighted by Reviewer LP6J). The model is established to be biologically grounded (concise reasons listed in lines 121-122), serving our purpose of building a neural computation testbed given its mechanistic nature and proper level of abstraction; other well-constructed mechanistic models of entorhinal-hippocampal interactions are not available to our best knowledge. Thus, we did not reinvent the wheel. However, the model’s
>
> 1. interaction with cortical and subcortical regions,
> 2. goal-directed task performance and neural representation under different neural computational rules,
> 3. application to neuroscience discoveries as an ML model,
>
> are unexplored—we explored all the above.
> The main contribution of our work is a biologically plausible ML-based framework (lines 77-82) that serves as a testbed of hypothesis-driven neural computation discoveries (Table 1, 2, Section 4). Further, we directly leveraged this framework to make straightforward, falsifiable neurophysiological predictions in important topics of neuroscience (Section 2.1). This aspect alone already makes our work well suited and important in our primary area, “**applications to neuroscience & cognitive science**.”

---

> ### Author Response · Authors · 2024-11-18
>
> *Thread [2/2]*
>
> **Regarding ML relevance:**
>
> > How can these findings be more directly linked to representation learning and machine learning at large?
>
> Neural representations, such as those of place cells and grid cells, are emerging properties of learning within biological neural networks; these representations can arise in artificial neural networks with appropriate biologically-imposed constraints as shown in our work. We study these representations using ML frameworks (lines 131-139). Our study on understanding how distinct cognitive processes like decision-making and spatial navigation are jointly possible in a limited number of interacting neural networks (brain regions) is of interest to both ML and Neuroscience communities. Navigation is very difficult for AI (Mirowski et al., ICLR 2017), and it’s even more so when they are combined with decision-making. Our understanding of what representations are needed to efficiently solve the combination using a limited network circuitry, with verifiability in both biological and artificial agents, is an important contribution. Further, our study matters for figuring out how artificial agents can truly make sense of their environments while being energy efficient, capable, and cognitively flexible, like humans and animals. This is related to achieving autonomous machine intelligence positioned and discussed by LeCun, 2022; we demonstrated, as a proof of concept, that sample-efficient learning in RL is achievable through an external content-addressable associative memory with a structured aspect. In particular, our work shows a structured conjunctive coding scheme (i.e., grid cells as a canonical example drawn from biological representation learning) is an important structural representation for forming cognitive maps (world models), enabling individuals to learn quickly and navigate spaces efficiently.
>
> We have revised our introduction with the above to emphasize the relevance of our work to ML (lines 65-70, in blue). Additionally, our work showcases ML techniques applied to neuroscience discoveries (our primary area of submission) for the reasons stated when addressing weaknesses. Thank you again for your feedback!
>
> > Section 2.3 is an odd aside, and the text does not seem to link it to the proposed model or findings at hand.
>
> We kindly disagree with the reviewer’s comment that Section 2.3 is an unrelated aside. The ML relevance of grid cell representation, which is the main subject of our study, is directly backed by the literature we mentioned in Section 2.3, e.g. Banino et al. demonstrated that artificial agents with grid cell-like representations have superior performance in navigation, a difficult ML task. Our findings provide additional insights, e.g., grid cell-like coding should also be efficient, i.e., utilizing all axes of the representation space (”conjunctive tuning”) in each module.
>
> **Regarding ethics concerns**
>
> Fig 1A is adapted from Chandra et al., 2023 with proper citation (line 179). Fig 1B is created by us, not from Chandra et al., 2023.
>
> ----
>
> Nieh et al., Geometry of abstract learned knowledge in the hippocampus. Nature 2021
>
> LeCun, Yann. "A path towards autonomous machine intelligence version 0.9. 2, 2022-06-27." Open Review 2022
>
> Banino et al., Vector-based navigation using grid-like representations in artificial agents, Nature 2018
>
> Mirowski et al., Learning to Navigation complex environments, ICLR 2017
>
> ----
> Please let us know if further clarifications are needed. Thank you again for your time and valuable feedback! We hope that our response provides sufficient reasons to raise the score.

---

### Author Response · Authors · 2024-11-25
**Kindly Reminder of Discussion Timeline & General Rebuttal Summary**

Dear reviewers,

Thank you again for your valuable time and constructive feedback!

We have uploaded a revised manuscript with edits in color after incorporating your valuable feedback. We have responded to each reviewer’s individual feedback, providing clarifications and highlighting relevant edits in each thread. Below, we summarize common questions and their corresponding manuscript changes, and other minor ones. We believe our paper has been significantly strengthened as a result of incorporating your feedback.

Given the last day of making any revision is this Wednesday AoE, we are keen to hear your thoughts and hope our revisions have sufficiently addressed your concerns and provided enough reasons for you to consider raising the score. Please let us know if we could assist with any further comments or questions.

Warm regards,

Authors of Submission 8300

----

**Summary of updates regarding common questions**
1. How grid code arises (Reviewers TiD8, bpUy, LP6J)
* New Fig 1C (& caption), lines 198-200, Appendix A.1.
2. Why not include CA3 recurrence (Reviewers bpUy, LP6J)
* Discussed in lines 516-525, with additional analysis of its impact in M2 & M4 in Appendix F.
3. More intuition on Fig 2 interpretation (Reviewers TiD8, LP6J, qnRf)
* Lines 357-359.

**Other minor edits in addressing questions by individual reviewers:**
1. ML relevance: lines 64-70 in Introduction.
2. Added clarity on model (hypothesis) difference: Fig 1E (& caption).
3. Modified RNN baselines (blue, black) in Fig 2.
3. Correct typo of M4 to M5: line 304.
4. Emphasize “rapid learning” is a relative term: emphasis added to lines 336 and 367.
5. Make Section 5.3.1 and Fig 5 caption more clear; add more quantification: reworded Fig 5 caption, added Fig 10.

---

### Author Response · Authors · 2024-12-02
**Kindly Reminder of Last Day of Discussion**

Dear reviewers,

Thank you very much for your valuable time and feedback! We are posting to kindly remind you that today is the last day reviewers may post a message. We are enthusiastic to hear from you if our responses have sufficiently addressed your concerns and provided grounds for your consideration of increasing the score. We’d also like to learn why if it’s not the case. Looking forward to hearing from you.

----

Warm regards,

Authors of submission 8300

---

### Meta-Review · Area_Chair_b2MW · 2024-12-24

**Metareview:**

This paper introduces a multi-region computational model integrating entorhinal and hippocampal dynamics with a reinforcement learning framework to study cognitive maps' role in decision-making. Using simulations of the accumulating tower task, the authors present results supporting the importance of conjunctive coding of spatial and sensory information in grid cells for rapid learning and efficient navigation. The study extends the Vector-HaSH framework to include sensory neocortex and action-selection RNN modules.

The reviewers found the study's goal and premise intriguing, emphasizing its potential contribution to computational neuroscience. However, significant concerns regarding its methodological execution, scope, and generalizability were raised. All but one reviewers voted for rejection.

**Additional Comments On Reviewer Discussion:**

An important criticisms raised by the reviewers and that was not resolved was that the study focuses on a single task without demonstrating generalizability to other environments or cognitive tasks, undermining the broader claims about "cognitive maps."

Other important comments included a) omission of CA3 recurrent dynamics may impact the validity of conclusions about conjunctive coding in the hippocampus, b) use of an RNN raises questions about whether the EC-HPC network or the RNN is responsible for temporal integration, and c) the model's learning is far from few-shot or one-shot learning, which contradicts common definitions of rapid learning in neuroscience. The authors responded to these comments, however reviewers were not fully convinced.

---

### Decision · Program_Chairs · 2025-01-22

Reject